# QUTE: Quantifying Uncertainty in TinyML models with Early-exit-assisted ensembles for model-monitoring

**Nikhil Pratap Ghanathe** [1]   **Steven J E Wilton** [1]

## Abstract

Uncertainty quantification (UQ) provides a resource-efficient solution for on-device monitoring of tinyML models deployed remotely without access to true labels. However, existing UQ methods impose significant memory and compute demands, making them impractical for ultra-low-power, KB-sized tinyML devices. Prior work has attempted to reduce overhead by using early-exit ensembles to quantify uncertainty in a single forward pass, but these approaches still carry prohibitive costs. To address this, we propose QUTE, a novel resource-efficient early-exit-assisted ensemble architecture optimized for tinyML models. QUTE introduces additional output blocks at the final exit of the base network, distilling early-exit knowledge into these blocks to form a diverse yet lightweight ensemble. We show that QUTE delivers superior uncertainty quality on tiny models, achieving comparable performance on larger models with **59% smaller model sizes** than the closest prior work. When deployed on a microcontroller, QUTE demonstrates a **31% reduction in latency** on average. In addition, we show that QUTE excels at detecting accuracy-drop events, outperforming all prior works.

## 1. Introduction

Recent advancements in embedded systems and machine learning (ML) have produced a new class of edge devices containing powerful ML models. These milliwatt-scale KB-sized devices, often termed *TinyML* devices, have low compute and memory requirements. They are often deployed in remote environments with no availability of true labels. This makes them susceptible to both out-of-distribution (OOD)

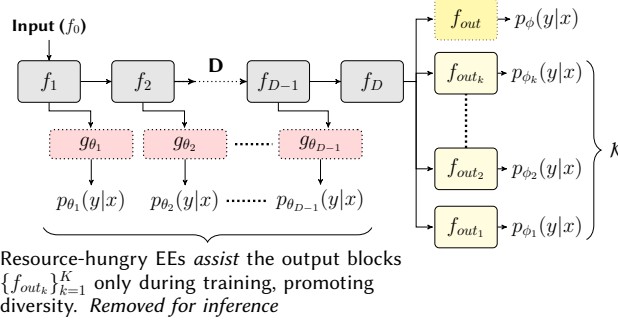

Figure 1: QUTE architecture. $\{f_{out}\}_{k=1}^{K}$ represents the 'K' additional output blocks at the final exit, which are *assisted* by 'K' early-exit blocks $\{g_{\theta_k}\}_{k=1}^{K}$ *only during training* to promote diversity (see Figure 2). For inference, all *early-exits* and $f_{out}$ are removed

and covariate-shifted data caused by environmental and sensor variations that manifest often unpredictably in the field. The ability of tinyML models to accurately measure the uncertainty of their predictions is crucial for two reasons. First, these devices generate data that is frequently used in critical decision-making. In the context of an autonomous vehicle (AV), uncertain predictions could result in the downstream system making cautious driving decisions (Tang et al., 2022). Second, tinyML devices often operate in remote environments (Vargas et al., 2021). If the inputs to the model changes (perhaps due to spatter, fog, frost, noise, motion blur, etc.), being aware of the model's unreliability may prompt an engineer to take remedial action. Many prior methods for drift/corruption detection are either statistical tests/methods that operate directly on input data (Bifet & Gavalda, 2007) with huge memory requirements, or require true labels for error-rate calculation (Gama et al., 2004; Baier et al., 2022). In contrast, uncertainty quantification (UQ) methods aim to estimate a model's *prediction confidence* without requiring true labels, providing an efficient solution for model monitoring, particularly in resource and power-constrained environments.

In ML models, there are two main types of uncertainty (Kendall & Gal, 2017): *epistemic* and *aleatoric*. Epis-

[1]Department of Electrical and Computer Engineering, University of British Columbia, Vancouver, Canada. Correspondence to: Nikhil Pratap Ghanathe <nikhilghanathe@ece.ubc.ca>.

*Proceedings of the $42^{nd}$ International Conference on Machine Learning*, Vancouver, Canada. PMLR 267, 2025. Copyright 2025 by the author(s).

temic uncertainty is *reducible* and stems from limited data or knowledge, while aleatoric uncertainty is *irreducible*, arising from inherent randomness in data or model limitations. In field-deployed tinyML models, both types of uncertainty can emerge due to semantic (OOD) and non-semantic changes in inputs. The more prevalent non-semantic category, which we term *corrupted-in-distribution* (CID) data, occurs when sensor malfunctions or environmental factors result in corrupted versions of expected data (e.g., fogged camera lens). CID data introduces both aleatoric (e.g., due to random noise or weather) and epistemic (e.g., from blur or digital corruptions) uncertainties, often simultaneously, making it crucial to capture these corruption-induced uncertainties to ensure model *reliability*. Unfortunately, modern neural networks are poor in estimating uncertainty of their predictions (Ovadia et al., 2019). Many prior works have proposed uncertainty-aware networks: ensemble networks (Lakshminarayanan et al., 2017) have been found to be effective in capturing both aleatoric and epistemic uncertainties. Alternatively, early-exit networks have been converted into ensembles (Antoran et al., 2020; Qendro et al., 2021; Ferianc & Rodrigues, 2023). However, both these approaches incur high memory/compute overheads and are untenable for tinyML.

In this paper, we propose QUTE (Figure 1), a novel and resource-efficient early-exit-assisted ensemble architecture that enables high-quality uncertainty quantification in tinyML models in the context of both in-distribution (ID) and corrupted-in-distribution (CID) data. As shown in Figure 1, we append additional lightweight classification heads to an existing base neural network to create ensemble members, and crucially, these classification heads are assisted by the early-exits using a novel distillation scheme (Ghanathe & Wilton, 2023) to promote diversity. Post-training, we eliminate the resource-hungry early-exits while retaining only the economical additional classification heads. Our approach has significantly less memory and compute overhead compared to prior works (59% smaller models and $3.2\times$ fewer FLOPS compared to the most relevant prior work (Qendro et al., 2021)). Furthermore, QUTE performs better than prior methods in estimating uncertainty caused due to CID error sources, and on-par with prior methods for uncertainty due to OOD. We further show that higher uncertainty is correlated with a drop in accuracy. We evaluate QUTE's ability to detect such accuracy drop events caused by CID against prior methods, and show that QUTE outperforms all prior methods. To the best of our knowledge, this is the first early-exit ensemble architecture for uncertainty quantification optimized for tinyML models.

This paper is organized as follows. Section 2 presents related work. The context and problem formulation are in Section 3. Our approach is described in Section 4. The experimental methodology and evaluation results are in Sections 5 and 6

respectively. Section 7 concludes the paper.

## 2. Related Work

**Uncertainty quantification** Bayesian neural networks (BNN) are well-suited to quantify uncertainty of a model (Blundell et al., 2015; Hernández-Lobato & Adams, 2015; Teye et al., 2018), but are parameter-inefficient and incur a high resource/compute overhead. Monte Carlo Dropout (MCD) (Gal & Ghahramani, 2016) creates implicit ensembles by enabling dropout during multiple inference passes. Recently, ensemble networks like Deep-Ensembles (Lakshminarayanan et al., 2017) have been shown to produce good uncertainty estimates (Zaidi & Zela, 2020; Wenzel et al., 2020; Rahaman et al., 2021). However, this too requires multiple inferences of individual networks, and is impractical in terms of memory for tinyML. Other prior works have proposed multi-input and multi-output networks that combine multiple independent networks into one (Havasi et al., 2021; Ferianc & Rodrigues, 2023), but they do not scale well and remain impractical for tinyML. (Ahmed et al., 2024) ensembles only normalization layers, but requires specialized hardware. Alternatively, prior works (Qendro et al., 2021; Ferianc & Rodrigues, 2023) have leveraged early-exit networks to create implicit ensembles. The closest work to QUTE is EE-ensemble (Qendro et al., 2021), which uses outputs of early-exits as ensemble members. However, the early-exits appended with extra learning layers incur a prohibitive cost (Section 3). In contrast, ensemble distillation methods (Havasi et al., 2021; Malinin et al., 2019; Tran et al., 2020) filter the knowledge of all ensemble members into a conventional neural network (NN). Hydra (Tran et al., 2020), which also uses a single multi-headed network is closest to our work. Appendix B.2 discusses other non-bayesian single-pass deterministic works (Van Amersfoort et al., 2020; Mukhoti et al., 2023; Sensoy et al., 2018; Deng et al., 2023) for UQ that are relevant for the resource-constrained tinyML space.

**Model monitoring** There has been a plethora of work to detect OOD samples, which represents a semantic shift from in-distribution (Yang et al., 2021; 2022; Zhang et al., 2023; Liu et al., 2020; Hendrycks & Gimpel, 2018; Dinari & Freifeld, 2022) . However, fewer works deal with detecting corrupted-ID, which represents a non-semantic shift w.r.t. ID, and are much harder to detect (Liang et al., 2020). Some prior works utilize corrupted-ID/covariate-shifted ID to better generalize on OOD (Bai et al., 2023; Katz-Samuels et al., 2022), which is difficult to obtain. The most relevant work that leverages *only* training data is generalized-ODIN (G-ODIN) (Hsu et al., 2020). It adds 1) a preprocessing layer that perturbs the input image and 2) decomposes confidence score for better OOD detection. However, the preprocessing

| Data Type | Model | Acc (↑) | ECE (↓) | BS (↓) | NLL (↓) |
|---|---|---|---|---|---|
| In-distribution | 1stack | 0.72 | 0.0343 | 0.0381 | 0.8174 |
| | 2stack | 0.78 | **0.0315** | 0.0305 | 0.6441 |
| | 3stack | 0.87 | 0.0373 | 0.0190 | 0.4063 |
| | 4stack | **0.89** | 0.0562 | **0.0161** | **0.4019** |
| Corrupted in-distribution | 1stack | 0.20 | 0.2796 | 0.0993 | 3.216 |
| | 2stack | 0.20 | **0.2518** | 0.0969 | **3.158** |
| | 3stack | **0.27** | 0.2534 | **0.0953** | 3.201 |
| | 4stack | 0.25 | 0.4255 | 0.1087 | 3.677 |

Table 1: Calibration metrics of Resnet with {1,2,3,4}-stack models on CIFAR-10. Top: in-distribution data. Bottom: corrupted in-distribution data. *Best results marked in bold.* For ECE, BS, NLL, *lower is better.*

might prove costly/impractical on tinyML devices. Recent studies (Xia & Bouganis, 2022; 2023; Jaeger et al., 2023) propose *failure detection* with a *reject* option, rejecting high-uncertainty samples from both ID (potentially incorrect predictions) and OOD, unlike traditional OOD detection which only separates ID from OOD.

**Early-exit networks** Early-exit networks add intermediate exits along the length of the base network thereby, providing avenues for reduction in average inference time (Teerapittayanon et al., 2016; Kaya et al., 2019; Huang et al., 2017; Bonato & Bouganis, 2021; Ghanathe & Wilton, 2023; Jazbec et al., 2024). Ghanathe & Wilton (2023) introduces an early-exit architecture optimized for tinyML models, and develops a knowledge distillation mechanism to mitigate network overthinking.

## 3. Background and Problem formulation

Consider an *in-distribution* dataset $S_{ID} = \{x_n, y_n\}_{n=1}^{N}$ of size $N$ where, $x_n$ and $y_n$ are the $n^{th}$ input sample and its corresponding true label respectively. A discriminative model $\mathcal{M}_\Theta(x)$ learns parameters $\Theta$ on $S_{ID}$ and outputs a class posterior probability vector $p_\Theta(y|x)$, which yields a predicted label $\hat{y}$. For a classification problem, $\hat{y} \in \{1, 2, ..L\}$, where $L$ is the number of classes. Uncertainty quantification methods endeavor to improve uncertainty estimation quality of $\mathcal{M}_\Theta(x)$ on $S_{ID}$. Specifically, we want to calibrate the model such that its *predictive confidence* ($\mathcal{C}_{\mathcal{M}_\Theta}$) is in sync with its accuracy ($\mathcal{A}_{\mathcal{M}_\Theta}$). A well-calibrated model will see a commensurate drop in $\mathcal{C}_{\mathcal{M}_\Theta}$ as $\mathcal{A}_{\mathcal{M}_\Theta}$ drops. The predictive confidence is given by, $\mathcal{C}_{\mathcal{M}_\Theta}(x) = \max_{l \in \{1,2,..,L\}} p_\Theta(y = l|x)$ Confidence-calibration is a well studied problem, and it helps achieve synergy between the predicted probabilities and ground truth correctness likelihood (Guo et al., 2017). However, when $\mathcal{M}_\Theta$ is deployed in a real-world scenario, it may encounter either 1) out-of-distribution data ($S_{OOD}$) or 2) a corrupted/covariate-shifted version ($S_{CID}$), due to environmental/sensor variations (E.g., frost, noise, fog, rain etc.). Unfortunately, we find that despite good uncertainty estimation quality on $S_{ID}$, many uncertainty-aware networks see a drop in quality and remain overconfident, particularly for

severely-corrupted $S_{CID}$ (Hsu et al., 2020). Interestingly, we find that the model becomes *less overconfident* to $S_{CID}$ as its *size shrinks*. This phenomenon is illustrated in Table 1. We report the calibration metrics of Resnet (He et al., 2016) on CIFAR10, with four model sizes ranging from 1 to 4 residual stacks on both $S_{ID}$ and $S_{CID}$. For calibration metrics (ECE, BS, NLL), *lower is better*. More details on calibration metrics and CIFAR10-CID can be found in Section 5 and Appendix D and A. As seen, the 4stack model with the highest model capacity is clearly the best on $S_{ID}$. However, on $S_{CID}$, it performs the worst, remaining overconfident in the presence of corruptions, even outperformed by 1stack. In contrast, 2stack and 3stack model are much better on $S_{CID}$. This demonstrates that smaller models are *less overconfident* for corruption in inputs, leading to better calibration on $S_{CID}$.

The capabilities of smaller models can be harnessed for better uncertainty estimation on $S_{CID}$ through early-exit networks, as illustrated by prior research like EE-ensemble (Qendro et al., 2021), which combines the predictions from multiple early-exits and final exit to form an ensemble. For example, in our evaluations, we find that on TinyImagenet-ID (Le & Yang, 2015) with MobilenetV2 (Howard et al., 2017), the negative log-likelihood (NLL) of EE-ensemble is *26% lower* than the popular Deep-Ensemble (Lakshminarayanan et al., 2017), and *34% lower* on TinyImagenet-Corrupted (Hendrycks & Dietterich, 2019) (lower is better). This showcases the crucial role of early-exits in achieving higher quality uncertainty overall. Further, these results align with prior findings (Hsu et al., 2020) that show that larger models overfit, leading to overconfidence, as they extract high-level abstractions that make distinguishing corrupted from clean inputs harder. In contrast, smaller models benefit from implicit regularization, preventing overfitting. They generalize better on corruptions by relying on more stable features instead of memorizing fine-grained ones. Our empirical results (Table 1) support this. From a Bayesian perspective, smaller models can exhibit a wider posterior due to implicit regularization, making them better at capturing uncertainty.

However, we find that EE-ensemble is resource-intensive because it adds additional learning layers at the early-exits to accommodate their varying learning capacities. For example, adding two early-exits *without additional learning layers* (say EE-0 and EE-1) after $1^{st}$ and $2^{nd}$ residual stack of a 3-residual stack Resnet for CIFAR10 results in individual accuracies of 0.602 and 0.707 at EE-0 and EE-1 respectively. This is significantly worse than the individual accuracy of the final exit, 0.84, which in turn leads to a poor ensemble behavior. To address this, EE-ensemble uses early-exits *with additional dense/fully-connected layers*, which incurs a high memory overhead affecting practicality in tinyML (Overhead$_{EE\text{-}ensemble}$ = 397K params compared to

Overhead$_{\text{QUTE}}$ = 8.2K params as shown in Section 4 & 6).

We present an alternative strategy to leverage the knowledge of early-exits in ensemble-creation that is extremely resource-efficient thereby enabling superior uncertainty estimates on both $S_{ID}$ and $S_{CID}$, which allows us to reliably detect accuracy-drops in the model due to $S_{CID}$.

## 4. QUTE

A neural network (NN) like the one shown in Figure 1, with depth $D$ is composed of several blocks of linear/non-linear operations (e.g., convolution, pooling) that are stacked. The NN consists of $D-1$ intermediate blocks $\{f_i(.)\}_{i=1}^{D-1}; f(.) = f_1(.) \circ f_2(.) \circ f_3(.)....\circ f_{D-1}(.) \circ f_D(.)$ and an output block $f_{out}(.)$. $f_0(.)$ is the input block. In the base network, the network processes the input $f_0(.)$ through each block $f_i(.)$ until layer $D$, ultimately producing prediction $p_\Theta(y|x)$ through output block $f_{out}(.)$.

Given a base NN (the grey blocks in Figure 1), we first create $K$ additional classification heads/output blocks $\{f_{out_k}\}_{k=1}^{K}$ at the final exit (after $f_D$) as shown in Figure 1. These output blocks constitute the ensemble $\mathcal{K}$. Next, we add $K$ early-exit blocks, $\{g_{\theta_k}\}_{k=1}^{K}$ along the length of the base network, which are used *only during training* for knowledge distillation. A more intuitive view of QUTE is presented in Figure 5. In Figure 1, $K$ is set to $D-1$. In practice, $K$ is a hyperparameter that depends on computation/resource budget and the required ensemble size $|\mathcal{K}|$, where the ensemble size is bounded above by the depth of the network. Thus, $\{f_{out_k}\}_{k=1}^{K}$ produces $K$ predictions, where each $f_{out_k}$ is assisted in producing predictions by its corresponding early-exit $g_{\theta_k}$ using *early-view assistance* method.

$$\mathcal{K} = \{p_{\phi_1}(y|x), p_{\phi_2}(y|x), p_{\phi_3}(y|x), ...., p_{\phi_K}(y|x)\} \quad (1)$$

Figure 2 illustrates the architecture of *early-view assistance*, where a single final exit block $f_{out_k}$ and its *assisting* early-exit $g_{\theta_k}$ is shown. As shown, we create an additional learning layer $h_{\phi_k}(.)$ at the $k^{\text{th}}$ output block. An identical learning layer $h_{\theta_k}(.)$ is also added at the $k^{\text{th}}$ early-exit. $h_{\phi_k}$ with parameters $\phi_k$ is responsible for assimilating knowledge from the corresponding $h_{\theta_k}$ with parameters $\theta_k$. $\sigma(.)$ is the output activation (E.g., dense+softmax). $h_{\phi_k}$ and $h_{\theta_k}$ are a single depthwise-convolution layer in our evaluations owing to its low computation and memory demand. We ensure equal dimensionality between $h_{\phi_k}$ and $h_{\theta_k}$ by adding a pointwise-convolution layer before $h_{\theta_k}$ at all early-exits (not shown in figure for brevity). Fundamentally, our aim is to imbue the diverse predictive behaviors of early-exits from different depths into the output blocks $\{f_{out_k}\}_{k=1}^{K}$.

**Training** Unlike EE-ensemble, all output blocks of QUTE have similar learning capacities since the input to all $\{h_{\phi_k}\}_{k=1}^{K}$ is $f_D$ of the base network. The output of $h_{\phi_k}$ is

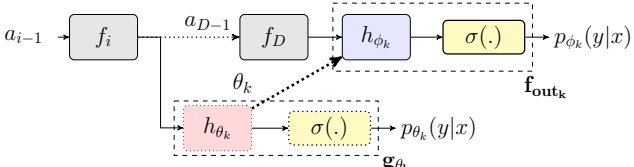

Figure 2: Early-view-assistance architecture. One *assisting* early-exit $g_{\theta_k}$ and corresponding *early-view-assisted* exit $f_{out_k}$ is shown. Early-exits weights $\theta_k$ are transferred/copied to $h_{\phi_k}$ before each train batch, i.e., $\phi_k = \theta_k$

termed as *early-view* since it is assisted by the early-exit. After attaching early-exits (EE) and early-view-assisted (EV) exits to the untrained base network, all weights are learned simultaneously during training. Unlike some prior works (Lakshminarayanan et al., 2017) that use extra data (e.g., adversarial samples), we only use the available training data. We use random data augmentation (E.g., random rotation/flip/crop), a well-known method to improve model robustness to corruptions to ensure fairness in accuracy-drop detection experiments (Section 6.2). The EEs are trained with a *weighted-loss* function, similar to previous studies (Kaya et al., 2019). Further, we assign a *higher weighting* factor, $w_{EV_k}$ for losses at EV-exits to minimize the effect of an overconfident $f_{out}$ (original output block). In addition, we increase each $w_{EV_k}$ by $\delta$ such that $w_{EV_k} = w_{EV_{k-1}} + \delta$ to vary the influence each EV-exit has on the weights of the shared base network, thereby promoting diversity. We empirically determine that $w_{EV_0} = 3$ and $\delta = 0.5$ yields the best results (see Appendix A for details). During training,

1) Before each train batch, we copy weights of $k^{\text{th}}$ EE-exit ($h_{\theta_k}$) into $k^{\text{th}}$ EV-exit ($h_{\phi_k}$) i.e., $\boldsymbol{\phi_k = \theta_k}$. Thus, each EV-exit iterates through a train batch with weights of the EE-exit. Weight transfer is limited solely from $h_{\theta_k}$ to $h_{\phi_k}$, with no other layers involved.

2) After a train batch, the loss computed at an EV-exit is w.r.t. the weights copied from EE, after which weights of $h_{\phi_k}$ are updated during backpropagation. However, before the next train batch begins, we overwrite the weights of $h_{\phi_k}$ by copying from EE. Therefore, the weights learnt by layers before/after $h_{\phi_k}$ always align with EE weights. Thus, this does not affect convergence (see Figure 6 and Section B.3).

$$\mathcal{L} = \sum_{k=1}^{K} \mathcal{L}_{EE_k} + \sum_{k=1}^{K} \mathcal{L}_{EV_k} + \mathcal{L}_{EF} \quad (2)$$

$$\mathcal{L}_{EE_k} = \tau_k \cdot \mathcal{L}_{CE} \qquad \mathcal{L}_{EV_k} = w_{EV_k} \cdot \mathcal{L}_{CE}$$

The training objective reduces to minimizing Eqn 2, where $\mathcal{L}$ is the total loss of QUTE, which is the sum of the losses of EE, EV and final exits. $\mathcal{L}_{CE}$ is the cross-entropy loss, and $\mathcal{L}_{EF} = \mathcal{L}_{CE}$, which is loss of original final exit. $\mathcal{L}_{EE_k}$ and $\mathcal{L}_{EV_k}$ are losses at the $k^{\text{th}}$ EE-exit and EV-exit respectively.

$\tau_k$ is the weighting applied to EE loss (Kaya et al., 2019). $w_{EV_k}$ is the weighting applied to the $k^{th}$ EV-exit.

The weights of base network (only grey blocks in Figure 1) are frozen for the last 10% of epochs while continuing weight transfer, enabling isolated training of EV-exits to foster increased diversity. In this way, the EE-knowledge is transferred to the final exit(s) to obtain better uncertainty estimation on $S_{ID}/S_{CID}$. In contrast, without EV-assistance, the various final exits risk learning the same predictive distribution, leading to poor ensemble behavior. We compare the effectiveness of EV-assistance in Appendix B.3.

**Inference** During inference, the resource-heavy EE-blocks are removed from the model, leaving only the base network and the lighter output blocks $\{f_{out_k}(.)\}_{k=1}^{K}$. We also eliminate the original output block $f_{out}$ due to its overconfident behavior, which compromises calibration quality. The final prediction ($p_\Theta(y|x)$) is obtained by calculating the mean of all $|\mathcal{K}|$ prediction vectors of $\mathcal{K}$.

$$p_\Theta(y|x) = \frac{1}{|\mathcal{K}|}\Big(\sum_{k=1}^{K} p_{\phi_k}(y|x)\Big) \qquad (3)$$

## 5. Evaluation Methodology

For our microcontroller (MCU) evaluations, we utilize two types of boards.

1) **Big-MCU**: The STM32 Nucleo-144 board (STM32F767ZI) (STMicroelectronics, 2019) with 2MB flash and 512KB SRAM, clocked at 216MHz with a typical power consumption of 285mW. This board accommodates high-resource baselines that require substantial memory and processing power.

2) **Small-MCU**: The STM32 Nucleo-32 board (STM32L432KC) (STMicroelectronics, 2018) with 256KB flash and 64KB RAM, clocked at 80MHz with a typical power consumption of 25mW. This ultra-low-power device better *reflects our goal of deployment* on smaller, energy-efficient systems.

We evaluate QUTE against several relevant baselines (Appendix A.1) in three settings: 1) Accuracy-drop/CID detection, which monitors system performance over time to detect gradual shifts and temporal degradation patterns (Section 6.2), 2) Failure detection, which targets instance-level identification of both misclassifications and OOD samples (Section 6.3), and 3) uncertainty estimation quality (Section 6.4). Since the focus of this work is to detect failures reliably in-the-field, accuracy-drop and failure detection experiments evaluate this capability.

**Datasets and Models** We evaluate QUTE on one audio classification and three image classification tasks of differing complexities: 1) MNIST (LeCun et al., 1998) on a 4-layer CNN, 2) SpeechCommands (Warden, 2018) on a 4-layer depthwise-separable model (DSCNN) (Zhang et al., 2017) for keyword spotting task, 3) CIFAR10 (Krizhevsky, 2009) on Resnet-8 (from MLPerf tiny benchmark suite (Banbury et al., 2021)) and 4) TinyImagenet (Le & Yang, 2015) on MobilenetV2 (Howard et al., 2017). While TinyImageNet/MobileNetV2 isn't a typical tinyML dataset/model, we include it to show QUTE's broader applicability. Despite their relative simplicity, these datasets exemplify the problem sizes often handled in KB-sized tinyML devices.

**CID datasets**: For CID datasets, we use corrupted versions of ID i.e., MNIST-C (15 corruptions) (Mu & Gilmer, 2019), CIFAR10-C (19 corruptions) and TinyImagenet-C (15 corruptions) (Hendrycks & Dietterich, 2019), with various corruptions such as frost, noise, fog, blur, pixelate etc. that induce both epistemic and aleatoric uncertainties (see Appendix A.2). For corrupting SpeechCmd, we use the audiomentations library (Jordal, 2024). We apply 11 types of corruptions such as noise, air absorption, time masking/stretching to obtain corrupted-SpeechCmd (denoted SpeechCmd-C). These corruptions affect the input data in various ways by altering key visual features (see Appendix A.2), potentially degrading model performance. For example, noise corruptions (e.g., impulse or gaussian) introduce random pixel-level variations, while weather-related corruptions (e.g., fog, snow, or rain) obscure critical visual elements necessary for accurate feature recognition.

**OOD datasets**: SpeechCommands has utterances of 35 words. Like Zhang et al. (2017), we train DSCNN to recognize ten words and use the rest as OOD (denoted SpeechCmd-OOD). Furthermore, we use FashionMNIST (Xiao et al., 2017) and SVHN (Netzer et al., 2011) as OOD for MNIST and CIFAR10 respectively. More details on models and datasets used can be found in Appendix A.

**Placement of Early-exits** The number of early-exits ($K$) is dictated by the resource budget. There are only a handful locations in a tinyML model where we can insert the early-exits. Our strategy is to insert them at equally-spaced locations along the base network. For MNIST and SpeechCmd, which both use 4-layer models, we insert two EEs after the $1^{st}$ and $2^{nd}$ layers. For Resnet-8 with 3 residual stacks, we insert EEs after $1^{st}$ and $2^{nd}$ residual stacks. We insert 5 EEs for MobilenetV2 at equally-spaced locations starting from the input.

**Accuracy-drop detection** Unlike with OOD data, correct predictions may still occur on CID inputs. Therefore, we focus on detecting accuracy-drop events caused by CID inputs. We present a realistic mechanism with no additional overhead that monitors *confidence* to detect a possible drop in accuracy. We formulate this as a binary classification problem. First, for each baseline we evaluate, we iterate over *only* ID and obtain model predictions while computing the moving average of accuracy of the past $m$ predictions using a sliding-window. The accuracy of the sliding-window is

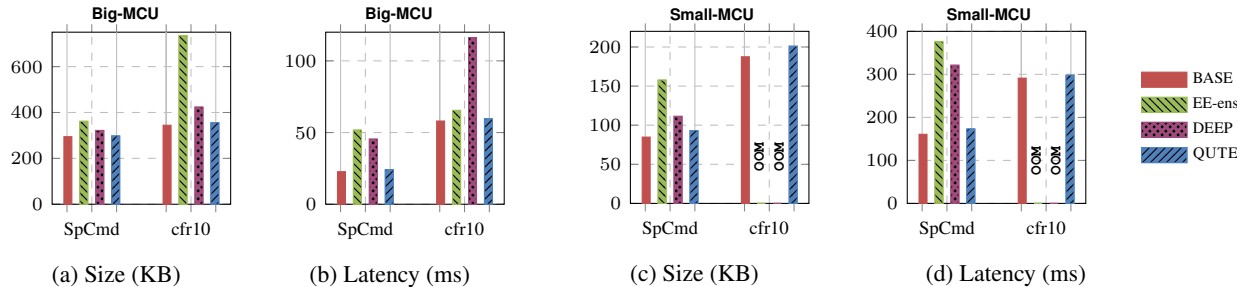

Figure 3: Microcontroller results for SpeechCmd (SpCmd), CIFAR10 (cfr10) on Big-MCU and Small-MCU (*lower is better*). EE-ensemble and DEEP on CIFAR10 **do not fit** i.e., out-of-memory (OOM)

denoted as $\mathcal{A}_{SW}$ and the accuracy distribution thus obtained is denoted as $\mathcal{A}_{ID}$ with mean $\mu_{ID}$ and standard-deviation $\sigma_{ID}$ respectively. In our empirical evaluations, we find that $m = 100$ is reasonable. We then construct corrupted datasets by appending ID with each corrupted-ID version. Next, we iterate over these ID+CID datasets (one for each type of corruption) using a sliding-window while computing the moving average of confidence ($\mathcal{C}_{SW}$). A *CID-detected* event corresponds to $\mathcal{C}_{SW} < \rho$, and a true positive event occurs when $\mathcal{C}_{SW} < \rho$ and $\mathcal{A}_{SW} \leq \mu_{ID} - 3 \cdot \sigma_{ID}$, where $\rho$ is a user-defined threshold. We vary $\rho$ from 0 to 1 in step sizes of 0.1 and repeat the whole experiment. Appendix A.6 provides additional details. Finally, we report the average area under precision-recall curve (AUPRC).

**Failure detection** Failure detection aims at rejecting both ID misclassifications and OOD instances, as both indicate a potential failure. To this end, we define two binary classification tasks: 1) ID✓ | ID×, which separates correct predictions from incorrect ones within ID/CID samples, and 2) ID✓ | OOD, which distinguishes correct predictions from OOD. For the former, we use ID+CID datasets where errors can arise from hard-to-classify ID samples or CID inputs, while for the latter, we use ID and OOD data. (Xia & Bouganis, 2023) leverages uncertainty estimates obtained through cascaded ensembles for failure detection, optimizing for a user-defined threshold. However, since this threshold may vary *in the field* depending on application requirements, we report the threshold-independent Area Under the Receiver Operating Characteristic curve (AUROC). Finally, Xia & Bouganis (2023) shows that Deep Ensembles (DEEP) achieve superior performance on threshold-free metrics, making DEEP the strongest baseline in our comparisons.

**Uncertainty quantification** For UQ, we report the class-weighted F1 score and two proper scoring metrics (Gneiting & Raftery, 2007): Brier score (BS) (Brier, 1950), which measures the accuracy of predicted probabilities and negative log-likelihood (NLL), which measures how close the predictions are to the ground truth (see Appendix D). Also, despite its documented unreliability in measuring uncertainty quality (Nixon et al., 2019), we report expected cali-

bration error (ECE) (Guo et al., 2017) due to it popularity. Appendix D discusses the limitations of ECE.

## 6. Results

The results section is organized as follows. First, Section 6.1 examines the MCU fit of QUTE compared to resource-heavy prior works, highlighting its **27% latency reduction** on Big-MCU and *29% smaller model size* compared to the lowest-latency baseline. On Small-MCU, high-resource prior methods **cannot even fit** when using CIFAR10, further underscoring QUTE's suitability for constrained environments. Next, Section 6.2 demonstrates QUTE's superiority over all baselines in accuracy-drop detection. On failure detection (Section 6.3), QUTE exceeds prior methods in distinguishing between ID✓ | ID× and is competitive on ID✓ | OOD with other baselines. Finally, Section 6.4 shows that QUTE exhibits good calibration quality especially on tiny-sized models with **59% fewer model parameters** on average compared to EE-ensemble. Additional results and ablation studies are presented in Appendix B.

### 6.1. MCU Fit: QUTE vs Resource-Heavy methods

We evaluate MCU implementations of QUTE on two MCUs of different sizes. Since the primary objective of TinyML systems is to minimize the *energy-per-prediction* (Ghanathe & Wilton, 2023), our target platform is the Small-MCU, as the base network fits comfortably on this smaller device.

We compare QUTE against the two most relevant baselines: EE-ensemble and DEEP. We exclude MCD due to its reliance on a specialized dropout module (Ahmed et al., 2023), which is impractical on MCUs. We also exclude HYDRA because it is often suboptimal, as shown in the following sections. Our evaluation uses CIFAR10 and SpeechCommands, as TinyImageNet/MobileNetV2 exceeds tinyML device limits. On Big-MCU, QUTE achieves **31% and 47% latency reductions** over EE-ensemble and DEEP, respectively, and maintains accuracy parity with both, even with **58% and 26% smaller models**. On Small-MCU, both EE-

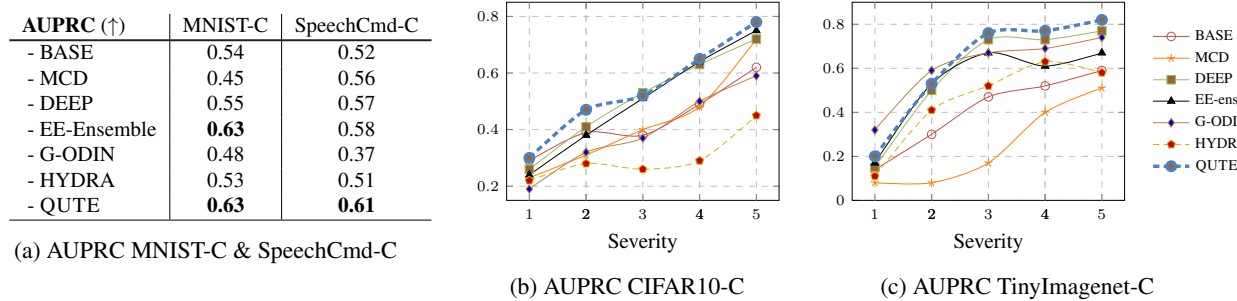

| AUPRC (↑) | MNIST-C | SpeechCmd-C |
|---|---|---|
| - BASE | 0.54 | 0.52 |
| - MCD | 0.45 | 0.56 |
| - DEEP | 0.55 | 0.57 |
| - EE-Ensemble | **0.63** | 0.58 |
| - G-ODIN | 0.48 | 0.37 |
| - HYDRA | 0.53 | 0.51 |
| - QUTE | **0.63** | **0.61** |

(a) AUPRC MNIST-C & SpeechCmd-C

(b) AUPRC CIFAR10-C

(c) AUPRC TinyImagenet-C

Figure 4: Accuracy-drop detection results. AUPRC is reported for all evaluated baselines (*higher is better*)

ensemble and DEEP *do not fit* on the device for CIFAR10; EE-ensemble model size exceeds Small-MCU's on-device memory by 270KB, whereas DEEP exceeds by 1.9KB. Notably, EE-ensemble has the highest peak RAM usage than DEEP and QUTE due to its need to store large intermediate feature maps during early-exit computation. In contrast, DEEP processes each ensemble member sequentially as there is no parallel compute capability on MCUs. This characteristic allows single-forward-pass methods like QUTE and EE-ensemble to achieve reduced latency, especially as the ensemble size increases. Appendix C delves into more details. These results highlight the memory limitations of ultra-low-power MCUs for tasks like model monitoring, emphasizing QUTE's practicality in such constrained settings where the *alternative may be no UQ or monitoring*.

### 6.2. Accuracy-drop detection

As described in Section 5, we aim at detecting accuracy-drops due to CID by monitoring the confidence. We also explore alternative uncertainty measures like predictive entropy. However, we choose confidence because it produces similar results without extra processing, a practical advantage for tinyML. Figure 4 reports AUPRC averaged over all corruptions for all datasets, and for five severity levels in the case of CIFAR10-C and TinyImagenet-C. By combining knowledge from EE at the final exits, QUTE consistently outperforms all baselines in detecting accuracy-drops caused due to CID across all datasets, which is crucial for maintaining model reliability in deployment.

MCD performs poorly in all cases either due to reduced model capacity (due to dropout) or due to extreme calibration. For example, on TinyImagenet-ID, MCD with an accuracy of 0.34 has a median confidence value of 0.345 across all predictions. However, on TinyImagenet-CID, the overparameterized nature of modern neural networks does not allow the median confidence of MCD to drop any further than 0.27 i.e., a drop of 0.075, which is not discernible enough to detect drop in accuracy. This is observed with EE-ensemble too for larger model sizes. In contrast, QUTE's median confidence on TinyImagenet-ID is 0.69 and the aver-

| AUROC | ID✓\|ID✗ | | | ID✓\|OOD | | |
|---|---|---|---|---|---|---|
| | Mnist | SpCmd | cfr10 | Mnist | SpCmd | cfr10 |
| -BASE | 0.75 | 0.90 | 0.84 | 0.07 | 0.90 | 0.88 |
| -MCD | 0.74 | 0.89 | **0.87** | 0.48 | 0.89 | 0.89 |
| -DEEP | 0.85 | **0.91** | 0.86 | 0.78 | **0.91** | 0.92 |
| -EE-ens | 0.85 | 0.90 | 0.85 | **0.85** | 0.90 | 0.90 |
| -G-ODIN | 0.72 | 0.74 | 0.83 | 0.4 | 0.74 | **0.95** |
| -HYDRA | 0.81 | 0.90 | 0.83 | 0.71 | 0.90 | 0.90 |
| -QUTE | **0.87** | **0.91** | 0.86 | 0.84 | **0.91** | 0.91 |

Table 2: Failure detection results

age median confidence on CID is 0.57, which helps create a much clearer distinction between ID and CID. EE-ensemble performs comparably to QUTE for tiny/medium-sized models. However, for large models, the additional learning layers added to balance the learning capacities of all exits cause the early-exits to function like conventional deep networks. Therefore, EE-ensemble becomes overconfident for severe corruptions, which results in a 8.9% drop in AUPRC on TinyImagenet-C. Similarly, G-ODIN remains overconfident on CID, and is surpassed by most baselines on all corrupted datasets. This suggests that general OOD-detectors are not automatically effective for CID detection, consistent with previous findings that OOD detectors find it hard to detect non-semantic shifts (Hsu et al., 2020). Further, HYDRA is consistently subpar at accuracy-drop detection, often being outperformed even by BASE. This highlights HYDRA's need for larger classification heads with more parameters to fully integrate ensemble-knowledge, consistent with the original paper's methodology. Contrarily, QUTE's early-exit distillation method with much fewer parameters (only a single depthwise CONV layer) proves to be more effective.

### 6.3. Failure detection

Table 2 reports the AUROC for failure detection experiments for ID✓| ID× and ID✓| OOD. As seen, for ID✓| ID×, QUTE outperforms all baselines on MNIST and SpeechCmd. MCD performs slightly better on CI-FAR10, likely due to the higher number of epistemic uncertainty-inducing corruptions present in CIFAR10-C, such as blur and digital distortions. This observation aligns

| Model | F1 ($\uparrow$) | BS ($\downarrow$) | NLL ($\downarrow$) | ECE ($\downarrow$) |
|---|---|---|---|---|
| **MNIST** | | | | |
| -BASE | 0.910±0.002 | 0.013±0.000 | 0.292±0.006 | 0.014±0.001 |
| -MCD | 0.886±0.004 | 0.018±0.000 | 0.382±0.004 | 0.071±0.006 |
| -DEEP | 0.931±0.005 | 0.010±0.000 | 0.227±0.002 | 0.034±0.004 |
| -EE-ensemble | 0.939±0.002 | 0.011±0.000 | 0.266±0.005 | 0.108±0.002 |
| -HYDRA | 0.932±0.006 | 0.010±0.000 | 0.230±0.012 | **0.014±0.005** |
| -QUTE | **0.941±0.004** | **0.009±0.000** | **0.199±0.010** | 0.026±0.003 |
| **SpeechCmd** | | | | |
| -BASE | 0.923±0.007 | 0.010±0.000 | 0.233±0.016 | 0.026±0.001 |
| -MCD | 0.917±0.006 | 0.011±0.000 | 0.279±0.013 | 0.048±0.002 |
| -DEEP | **0.934±0.008** | 0.008±0.000 | 0.205±0.012 | 0.034±0.006 |
| -EE-ensemble | 0.926±0.002 | 0.009±0.000 | 0.226±0.009 | 0.029±0.001 |
| -HYDRA | 0.932±0.005 | 0.008±0.000 | **0.203±0.016** | 0.018±0.004 |
| -QUTE | 0.933±0.006 | **0.008±0.000** | 0.202±0.016 | **0.018±0.001** |
| **CIFAR10** | | | | |
| -BASE | 0.834±0.005 | 0.023±0.000 | 0.523±0.016 | 0.049±0.003 |
| -MCD | 0.867±0.002 | 0.019±0.000 | 0.396±0.003 | 0.017±0.005 |
| -DEEP | 0.877±0.003 | 0.017±0.000 | **0.365±0.015** | **0.015±0.003** |
| -EE-ensemble | 0.854±0.001 | 0.021±0.000 | 0.446±0.011 | 0.033±0.001 |
| -HYDRA | 0.818±0.004 | 0.026±0.000 | 0.632±0.017 | 0.069±0.001 |
| -QUTE | 0.858±0.001 | 0.020±0.000 | 0.428±0.019 | 0.025±0.003 |
| -QUTE + | **0.878±0.003** | **0.017±0.000** | 0.369±0.008 | 0.026±0.001 |
| **TinyImagenet** | | | | |
| -BASE | 0.351±0.005 | 0.004±0.000 | 5.337±0.084 | 0.416±0.003 |
| -MCD | 0.332±0.004 | 0.003±0.000 | 2.844±0.028 | 0.061±0.005 |
| -DEEP | 0.414±0.006 | 0.003±0.000 | 3.440±0.049 | 0.115±0.003 |
| -EE-ensemble | **0.430±0.005** | 0.003±0.000 | 2.534±0.046 | **0.032±0.006** |
| -HYDRA | 0.376±0.004 | 0.004±0.000 | 3.964±0.036 | 0.328±0.004 |
| -QUTE | 0.395±0.014 | 0.004±0.000 | 3.700±0.123 | 0.282±0.009 |
| -QUTE + | 0.381±0.010 | **0.003±0.000** | 2.757±0.044 | 0.122±0.008 |

Table 3: Calibration Metrics for all baselines on ID data. The best results are marked in bold. All results are mean ± std-dev for three independent splits of ID test data.

with prior work highlighting MCD's strength in capturing instance-level epistemic uncertainties (Kendall & Gal, 2017). For ID✓| OOD, QUTE exceeds all baselines on SpeechCmd, and is a close second on MNIST. Surprisingly, G-ODIN, a specialized OOD detector, is poor on MNIST and SpeechCmd, even outperformed by BASE. Further analysis revealed that G-ODIN is under-confident on these ID-datasets and over-confident on OOD. As an ablation study, we increased the number of training epochs of G-ODIN by 30 epochs, which resulted in a 55% improvement on ID✓| OOD on MNIST. Contrarily, on a larger model, Resnet-8 on CIFAR10, G-ODIN's performance notably improves on ID✓| OOD, consistent with previous findings. This may suggest the need to rethink OOD detection with extremely tiny models. These results showcase QUTE's versatility/efficacy in detecting both CID and OOD data.

### 6.4. Uncertainty quantification

QUTE consistently matches or even *outperforms* all baselines on tiny-sized models such as MNIST and SpeechCmd, and remains *competitive* on medium and large-scale benchmarks like CIFAR-10 and TinyImageNet—despite operating with significantly fewer computational resources. All methods outperform BASE, except MCD on MNIST and SpeechCmd, where inference-time dropout reduces model capacity, causing accuracy drops and poor calibration. DEEP showcases good calibration on all datasets owing to its larger model capacity. However, MCD and DEEP are

impractical for tinyML because they both require multiple inference passes or specialized hardware.

**QUTE with relaxed resource constraints**: QUTE also outperforms EE-ensemble, the most relevant prior work, across all datasets with models that are, on average, 59% smaller, except on larger datasets and models like TinyImagenet on MobileNetV2. Since QUTE's primary focus is resource efficiency, the limited parameters in its output blocks—restricted to a single depthwise convolutional (DCONV) layer—can constrain calibration performance relative to high-resource methods like DEEP and EE-ensemble. To normalize the comparison, we enhanced the QUTE architecture with additional learning layers at each output block, creating QUTE +. For Resnet-8 on CIFAR10, we added two depthwise-separable CONV layers, while for MobileNetV2 on TinyImageNet, we included an additional dense (fully connected) layer to form QUTE +. As seen in Table 3, F1 score of QUTE + exceeds that of DEEP on CIFAR10, however, it slightly degrades on TinyImagenet. Further analysis revealed that since the base network is trained simultaneously with all early-exits (EE) and EV-exits, the weights of the shared base network is negatively impacted since the training routine tries to optimize for all exits. In contrast, EE-ensemble method employs only half as many exits (only EEs) compared to QUTE, and the EE's regularization effect helps network accuracy (Teerapittayanon et al., 2016). Appendix B.4 studies the effect of ensemble size on QUTE's calibration performance. Nevertheless, QUTE + significantly improves calibration, outperforming all baselines on BS and achieving NLL on par with the top-performing method. These results indicate that, in an unconstrained setting, QUTE can achieve calibration performance comparable to high-resource methods.

Additionally, we evaluate QUTE's early-exit distillation against HYDRA's ensemble-distillation. While HYDRA's lightweight heads capture ensemble knowledge in tiny models, they falter with larger models, reducing accuracy and calibration, even underperforming BASE on CIFAR10. This highlights HYDRA's dependency on larger heads for effective ensemble distillation.

## 7. Conclusion and Discussion

Uncertainty quantification is essential for model monitoring in tinyML, yet many prior works fail to address the resource constraints inherent in this domain. We propose a novel resource-efficient ensemble architecture for uncertainty estimation that is optimized for tinyML. At the core of our methodology is our finding that model overconfidence decreases with its size. We leverage the better uncertainty estimation quality of early-exits (EE) by injecting EE weights into the multiple lightweight classification heads created at the output. QUTE provides reliable uncertainty esti-

mates in a single forward pass with **59% smaller models** on average and a **31% average reduction in latency** on low-power MCUs. Furthermore, QUTE effectively distills EE knowledge into the final exits, enabling it to capture both aleatoric and epistemic uncertainties more effectively. This capability leads to superior performance in detecting accuracy-drop events (corruptions) across a diverse range of datasets and model complexities. QUTE serves as a practical and cost-effective *accuracy-monitoring* mechanism for field-deployed models.

*Limitations/Discussions* The ensemble size ($K$) is restricted by the depth of the base network, which may not provide uncertainty estimates of sufficient quality for a given safety-critical application. In addition, we find that increasing $K$ beyond a certain point negatively influences the shared base network, leading to degradation in calibration (ablation study in Appendix B.4). Furthermore, while QUTE achieves the best calibration on tiny-sized models under severe resource constraints, it requires additional learning layers on large models and datasets. Alternatively, methods that improve early-exit calibration (Jazbec et al., 2024) could potentially improve effectiveness of QUTE's EV-assistance. Interestingly, we observe that excessively increasing EV-assistance, such as by raising the depth multiplier in QUTE 's output blocks, negatively impacts accuracy despite improving calibration. However, its worth noting that the training times with QUTE are higher compared to other methods (except HYDRA) due to weight transfer between train-batches. Finally, despite QUTE's versatility/effectiveness in OOD detection, specialized OOD detectors should be considered, especially for large models.

## Acknowledgments

NSERC and COHESA support gratefully acknowledged.

## Impact Statement

This work presents QUTE, a framework for efficient uncertainty quantification and model monitoring in field-deployed TinyML models, pushing the boundaries of resource-constrained machine learning. QUTE supports reliable, real-time decision-making in resource-constrained, high-stakes scenarios. While the work primarily aims to improve TinyML reliability, care must be taken to ensure its responsible use in safety-critical applications. In an era of abundant compute, where conventional neural networks often operate inefficiently (Patterson et al., 2021), QUTE exemplifies the potential of energy-efficient strategies to reduce unnecessary energy costs.

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

# Appendix

## A. Training and dataset details

In this section, we describe the details of the models evaluated and specifics of training and the experimental setup.

### A.1. BASELINES

**BASE**: Unmodified implementations of models evaluated.

**Monte Carlo Dropout (MCD)** (Gal & Ghahramani, 2016): computes mean of prediction vectors from $K$ inference passes by activating *dropout*. For fairness, dropout and EE locations are same. We use a dropout rate of 0.1, 0.05, 0.2, 0.2 for MNIST, SpeechCmd, CIFAR10 and TinyImagenet respectively.

**Deep Ensembles (DEEP)** (Lakshminarayanan et al., 2017): explicit ensemble of models with same base model architecture but with different random weight initializations. The number of ensemble members = $K$.

**Early-exit Ensembles (EE-ensemble)** (Qendro et al., 2021): attaches multiple early-exits with additional fully-connected (FC) layers. The size of the FC layers is chosen such that the model delivers the best uncertainty estimation. All early-exit prediction vectors including that of the final exit are averaged to get the final ensemble prediction. We place the early-exits at the same locations as QUTE. Ablation studies with EE-ensemble are presented in Appendix B.5.

**HYDRA** (Tran et al., 2020): uses the same architecture as QUTE. Instead of early-exit distillation like QUTE, it employs ensemble distillation, capturing unique predictive behavior of individual ensemble members. We include this baseline to assess QUTE's early-exit distillation method against ensemble distillation approaches.

**Generalized-ODIN (G-ODIN)** (Hsu et al., 2020): a generalized OOD detection method that decomposes the confidence score and introduces a preprocessing layer that perturbs the inputs for better OOD detection. The perturbation magnitude is determined on a small held-out validation set.

### A.2. Datasets

In our evaluations, we use four in-distribution datasets for training all baselines methods we evaluate: 1) MNIST (LeCun et al., 1998), 2) SpeechCommands (Warden, 2018), 3) CIFAR10 (Krizhevsky, 2009) and 4) TinyImagent (Le & Yang, 2015).

**MNIST**  MNIST is a dataset of handwritten digits containing 60,000 grayscale images of size 28×28 and a test set of 10,000 images. MNIST contains 10 classes. Although a simple dataset, MNIST reflects the problem sizes typical of many tinyML applications.

**SpeechCommands**  Speech-Cmd is a collection of short audio clips, each spanning 1 second. The dataset consists of utterances for 35 words and is commonly used for benchmarking keyword spotting systems. We train our model (DSCNN) to recognize ten words out of 35: *Down, Go, Left, No, Off, On, Right, Stop, Up, Yes*. Thus, the number of classes is 10. The audio files in WAV format are preprocessed to compute Mel-frequency cepstral coefficients (mel-spectograms). The mel-spectograms are of size 49×10 with a single channel.

**CIFAR10**  CIFAR10 dataset consists of 60,000 32×32 rgb images out of which 10,000 images are in the test set. It contains 10 classes and thus 6000 images per class.

**TinyImagenet**  TinyImagenet is a smaller version of the Imagenet (Deng et al., 2009) dataset containing 200 classes instead of 1000 classes of the original Imagenet. Each class in TinyImagenet has 500 images in the train set and the validation set contains 50 images per class. The size of the images are resized and fixed at 64×64×3.

#### A.2.1. CORRUPTED DATASETS

For image classification tasks, we use the following corrupted versions of ID: 1) MNIST-C (Mu & Gilmer, 2019), 2) CIFAR10-C and 3) TinyImagenet-C (Hendrycks & Dietterich, 2019). All corruptions are drawn from 4 major sources: noise, blur, weather and digital. The *digital* and *blur* corruptions induce high epistemic uncertainty, whereas, *noise* corruptions induce higher aleatoric uncertainty. In contrast, *weather* corruptions tend to encompass both epistemic and aleatoric components. Most corruptions are systematic transformations of input images, which predominantly introduce epistemic uncertainty. Pure noise corruptions are the primary source of aleatoric uncertainty; however, many corruptions exhibit both

components, often leaning toward epistemic.

**MNIST-C** The MNIST-C dataset contains 15 corrupted versions of MNIST - *shot_noise, impulse_noise, glass_blur, fog, spatter, dotted_line, zigzag, canny_edges, motion_blur, shear, scale, rotate, brightness, translate, stripe, identity*. All the corruptions are of a fixed severity level.

**CIFAR10-C** CIFAR10-C includes 19 different types of corruptions with 5 severity levels which gives us 19×5=95 corrupted versions of CIFAR-10 ID. We use all 95 corrupted versions in our experiments. The list of corruptions are: *gaussian_noise, brightness, contrast, defocus_blur, elastic, fog, frost, frosted_glass_blur, gaussian_blur, impulse_noise, jpeg_compression, motion_blur, pixelate, saturate, shot_noise, snow, spatter, speckle_noise, zoom_blur*.

**TinyImagenet-C** TinyImagenet-C includes 15 different types of corruptions with 5 severity levels which gives us 15×5=75 corrupted versions of ID. We use all 75 corrupted versions in our experiments. The list of corruptions are: *gaussian_noise, brightness, contrast, defocus_blur, elastic_transform, fog, frost, glass_blur, impulse_noise, jpeg_compression, motion_blur, pixelate, shot_noise, snow, zoom_blur*.

The above corruptions distort key visual features of the input, as outlined below.

- **Noise corruptions** (e.g., gaussian/shot/impulse/speckle noise) introduce random pixel variations, which disrupts the model's pattern recognition capability.

- **Blur corruptions** (e.g., defocus/motion/gaussian/zoom/frosted glass blur) distort the edges and textures of the objects in the images, affecting spatial feature extraction.

- **Digital corruptions** (e.g., JPEG compression artifacts, pixelation, contrast shifts, saturation) modify pixel distributions, which impact feature consistency.

- **Weather corruptions** (e.g., fog, frost, snow, rain) obscure key visual components important for feature recognition, reducing visibility and contrast.

**SpeechCmd-C** For the audio classification task of Keyword spotting on SpeechCmd, we corrupt the audio before converting them into mel-spectrograms using the audiomentations library (Jordal, 2024) with the following 11 corruptions: *gaussian noise, air absorption, band pass filter, band stop filter, high pass filter, high shelf filter, low pass filter, low shelf filter, peaking filter, tanh distortion, time mask, time stretch*.

### A.3. Training details

We evaluate four models in our experiments:1) 4-layer CNN on MNIST, 2) 4 -layer Depthwise-separable CNN (DSCNN) on SpeechCmd, 3) Resnet-8 with 3 residual stacks from the MLPerf Tiny benchmark suite (Banbury et al., 2021) on CIFAR10 and 4) MobilenetV2 (Howard et al., 2017) on TinyImagenet.

The 4-layer CNN is trained for 20 epochs with a batch size of 256, the DSCNN model is trained for 10 epochs with a batch size of 100, the Resnet-8 model is trained for 200 epochs with batch size of 32 and the MobilenetV2 model is trained for 200 epochs with a batch size of 128. All models are trained with Adam optimizer with momentum of 0.9 and an initial learning rate of 0.001, expect DSCNN on SpeechCmd which uses an initial learning rate of 0.0005. The learning rate is decayed by a factor of 0.99 every epoch for all image classification tasks. We follow a step function for SpeechCmd that reduces learning rate by half every 2 epochs. For models with QUTE architecture, we transfer weights from early-exits to all $\{f_{out}(.)\}_{k=1}^{K}$ using a callback that sets the weights of each $f_{out_k}$ with weights copied from the corresponding assisting early-exit at the beginning of each training batch.

### A.4. Weighting the loss at EV-exits

As described in Section 4, we weight the loss at EV-exits with a higher factor $w_{EV_k}$ such that $w_{EV_k} = w_{EV_{k-1}} + \delta$. This is done to further promote diversity across the EV-exits and to vary the influence each EV-exit has on the shared network's weights. Utilizing distinct weighting factors across EV-exits implies that each EV-exit contributes to the final loss calculation with varying degrees of influence. Consequently, this diversity extends to the weight updates of the shared base network, thereby injecting diversity into the classification heads. We empirically set $\delta$ to be 0.5. To determine $w_{EV_0}$, we repeat training with $w_{EV_0}$ set to $\{2, 3, 4, 5\}$. We find that for $w_{EV_0} > 3$, the NLL starts dropping steadily because the higher

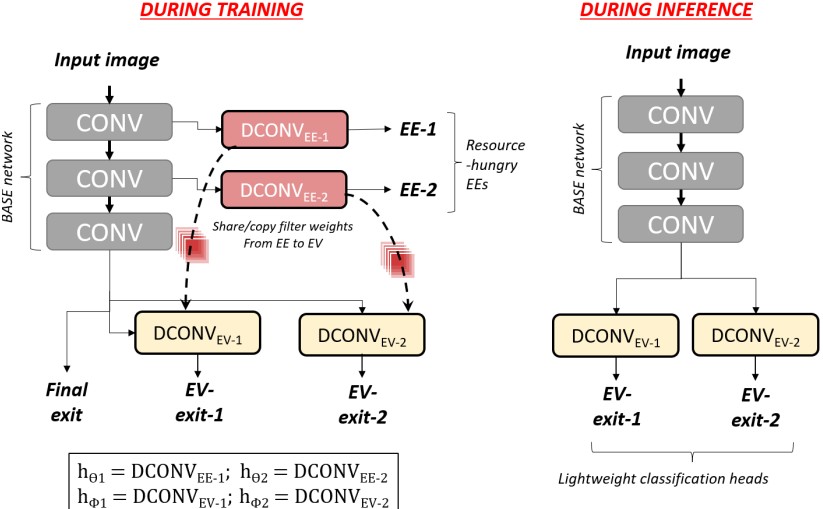

Figure 5: Depiction of QUTE during training and inference. The early-exits (EE) are only used during training for weight-transfer. During inference, the EEs are removed, and only EV-exits are retained. We do not include other layers such as dense and pointwise convolution in the figure for brevity.

loss at EV-exits overshadow the losses of the early-exits. This diminishes the influence of early-exits at EV-exits, which is undesirable. Hence, we set $w_{EV_0} = 3$ in all our experiments.

### A.5. Corrupted-in-distribution datasets

We construct the corrupted-in-distribution (CID) datasets for our evaluations as follows. For MNIST-C with a fixed severity level, we append each corrupted version of MNIST to the MNIST-ID dataset to create 15 ID+CID datasets. For CIFAR10-C and TinyImagenet-C, which have 19 and 15 different corruptions respectively with 5 severity levels each, we construct the corrupted datasets as follows. For each type of corruption, we randomly select $p$ samples from each of the 5 severity levels. Next, we concatenate all these samples to create a new corrupted dataset of size $5 \times p$. $p$ is selected such that $5 \times p$ = size of ID dataset. This process is repeated for all corruptions. The datasets thus obtained contain samples from all severity levels. For example, for CIFAR10-C which consists of 19 corruptions, this process yields 19 ID+CID datasets.

### A.6. Accuracy-drop detection experiments

For experiments to detect accuracy drop/CID detection described in Section 6.2, we append the ID dataset with each corrupted dataset obtained from the methodology described above. For example, for experiments with CIFAR10-C, we obtain 19 ID+CID datasets that contains both ID samples and corrupted samples (from all severity levels). Next, as described in Section 5, for each prior work we evaluate, we first iterate over *only* ID and obtain predictions from the model while computing the moving average of accuracy of the past $m$ predictions using a sliding-window. In this way, we obtain the accuracy distribution of the sliding-window $\mathcal{A}_{ID}$ on ID, and then compute its mean $\mu_{ID}$ and standard-deviation $\sigma_{ID}$. The accuracy of the sliding-window at any given point of time is denoted as $\mathcal{A}_{SW}$. Next, we iterate over all ID+CID datasets while computing the moving average of confidence of the past $m$ predictions using a sliding-window ($\mathcal{C}_{SW}$). We record all instances of $\mathcal{C}_{SW}$ dropping below a certain threshold $\rho$. These events are denoted as *CID-predicted* events. At the same time, we also compute $\mathcal{A}_{SW}$ for evaluating how many CID-predicted events are actually a *CID-detected* event (true positive). Finally, the description of true positive (TP), false positives (FP), false negatives (FN) and true negatives (TN) are as follows.

- True positive: $\mathcal{C}_{SW} < \rho$ and $\mathcal{A}_{SW} \leq \mu_{ID} - 3 \cdot \sigma_{ID}$

- False positive: $\mathcal{C}_{SW} < \rho$ and $\mathcal{A}_{SW} > \mu_{ID} - 3 \cdot \sigma_{ID}$

- True negative: $\mathcal{C}_{SW} > \rho$ and $\mathcal{A}_{SW} > \mu_{ID} - 3 \cdot \sigma_{ID}$

- False negative: $\mathcal{C}_{SW} > \rho$ and $\mathcal{A}_{SW} \leq \mu_{ID} - 3 \cdot \sigma_{ID}$

| Model | F1 (↑) | BS (↓) | NLL (↓) |
|---|---|---|---|
| **MNIST** | | | |
| - BASE-TS | 0.937±0.002 | 0.009±0.000 | 0.203±0.005 |
| - QUTE-TS | **0.942±0.001** | **0.008±0.000** | **0.191±0.005** |
| - QUTE | 0.941±0.004 | 0.009±0.000 | 0.199±0.010 |
| **SpeechCmd** | | | |
| - BASE-TS | 0.923±0.007 | 0.009±0.000 | 0.229±0.017 |
| - QUTE-TS | 0.926±0.006 | 0.009±0.000 | 0.233±0.010 |
| - QUTE | **0.933±0.006** | **0.008±0.000** | **0.202±0.016** |
| **CIFAR10** | | | |
| - BASE-TS | 0.834±0.000 | 0.023±0.000 | 0.493±0.000 |
| - QUTE-TS | 0.852±0.002 | 0.021±0.000 | 0.444±0.013 |
| - QUTE | **0.858±0.001** | **0.020±0.000** | **0.428±0.019** |

Table 4: Calibration with temperature scaling (TS) on ID

| AUPRC (↑) | MNIST-C | SpeechCmd-C | CIFAR10-C (Severity 1 to 5) |
|---|---|---|---|
| - BASE-TS | 0.47 | 0.52 | 0.30, 0.42, 0.38, 0.50, 0.61 |
| - QUTE-TS | 0.52 | 0.53 | 0.26, 0.45, 0.51, 0.65, 0.76 |
| - QUTE | **0.63** | **0.61** | **0.30, 0.47, 0.52, 0.65, 0.78** |

Table 5: Accuracy-drop detection with temperature-scaling

In this way, we collect the TP, FP, TN, FN from each ID+CID datasets and compute the average precision and recall for a given threshold $\rho$. Finally, we report the AUPRC. We choose AUPRC because the precision-recall curve is resistant to imbalance in datasets.

## B. Ablation Studies and Additional Results

### B.1. Comparison with Temperature scaling

Temperature scaling (TS) (Guo et al., 2017) is a simple post-training calibration technique that applies a linear scaling factor to logits (pre-softmax) to mitigate model overconfidence. Table 4 and Table 5 present the calibration and accuracy-drop detection results, respectively, for TS versions of BASE (BASE-TS) and QUTE (QUTE-TS). For QUTE-TS, the same architecture as QUTE is used, but without weight transfer. We employed the best-performing *pool-then-calibrate* configuration from (Rahaman et al., 2021), which applies temperature scaling to ensembles to obtain QUTE-TS. As the results show, QUTE consistently outperforms BASE-TS on both calibration and accuracy-drop detection. However, despite improving calibration in some scenarios, QUTE-TS performs worse than QUTE on accuracy-drop detection, particularly with tiny datasets such as MNIST-C and SpeechCmd-C. A deeper analysis revealed that QUTE-TS tends to be overconfident for certain corruptions. For instance, under fog corruption on MNIST-C, QUTE-TS exhibits an overconfident behavior with a median confidence of 0.99, while QUTE maintains a much lower confidence of 0.51, leading to superior accuracy-drop detection capability.

### B.2. Comparison with single-pass deterministic methods

**Related work**: Several single-pass (non-Bayesian) methods for UQ (Van Amersfoort et al., 2020; Mukhoti et al., 2023; Sensoy et al., 2018; Deng et al., 2023) offer lower memory footprints compared to ensemble methods. However, they remain impractical for tinyML due to specialized output layers or architectural constraints. DUQ (Van Amersfoort et al., 2020) adds a specialized layer post-softmax, drastically increasing resource use (e.g., 10× more parameters for Resnet on CIFAR10). DDU (Mukhoti et al., 2023) simplifies this but relies on residual connections for feature space regularization, limiting its applicability. Priornets (Malinin & Gales, 2018) require OOD data, which is often unrealistic. Postnets (Charpentier et al., 2020), though OOD-free, focuses primarily on OOD detection, often sacrificing accuracy below the base network's level. Other methods like (Meronen et al., 2024) that address overconfidence in early-exit networks require excessive computational resources (e.g., laplace approximations).

**Comparison with Postnets** Postnets (PostN) (Charpentier et al., 2020) is the most relevant single-pass deterministic method compared to QUTE, which uses normalizing flows to predict posterior distribution of any input sample with no additional

| Model | F1 (↑) | BS (↓) | NLL (↓) |
|---|---|---|---|
| MNIST-PostN | 0.92 | 0.012 | 0.286 |
| MNIST-QUTE | **0.94** | **0.009** | **0.199** |
| CIFAR-PostN | 0.84 | 0.022 | 0.462 |
| CIFAR-QUTE | **0.858** | **0.020** | **0.428** |

Table 6: Calibration comparison with Postnets on ID

| Model | F1 (↑) | | BS (↓) | | NLL (↓) | |
|---|---|---|---|---|---|---|
| | EV | no-EV | EV | no-EV | EV | no-EV |
| CIFAR10 | 0.858±0.001 | 0.858±0.000 | **0.0205±0.000** | 0.0206±0.000 | **0.435±0.007** | 0.459±0.006 |
| TinyImagenet | **0.380±0.011** | 0.35±0.004 | **0.0043±7.2E-5** | 0.0046±5.7E-5 | **3.813±0.105** | 4.743±0.111 |

Table 7: Effectiveness of *early-view* assistance. Results averaged over 3 independent training runs.

memory overhead. We evaluate PostN on MNIST and CIFAR10 by substituting the encoder network architectures with the ones used in our evaluations. We use identical hyperparameter settings of Charpentier et al. (2020) and train for same number of epochs as in our evaluations but with early stopping. Table 6 reports the calibration metrics for PostN and QUTE. Since PostN don't necessarily focus on accuracy, we found that MNIST-PostN needed a 50% increase in number of epochs to even surpass the accuracy of MNIST-BASE. We report this result. For CIFAR10, we found that early stopping halted the training, and report those results. As seen, QUTE consistently outperforms PostN on all metrics on both datasets, making QUTE a more efficient choice for resource-constrained environments.

### B.3. Effectiveness of EV-assistance method and its effect on convergence

To investigate the effect of the early-view (EV) assistance method on ensemble quality, we train non-EV versions of QUTE i.e., the same architecture (with early-exits) except without the weight transfer mechanism. Table 7 reports the calibration metrics for both EV and non-EV versions for CIFAR10 and TinyImagenet. As shown, EV-assistance has a massive influence on improving calibration, and improves the network accuracy on TinyImagenet by 8%. Also, we find that the ability of non-EV versions to detect accuracy-drop events reduces by 15% on average compared to EV-versions. These results show that the notably lower NLL obtained with EV-assistance (5% lower on CIFAR10 and 19% lower on TinyImagenet) not only improves uncertainty estimates, but it also proves crucial in both CID and OOD detection. Furthermore, we conducted an ablation study to isolate the impact of early-exit (EE) knowledge transfer on ensemble diversity. We quantify QUTE's ensemble diversity using the normalized disagreement (ND) metric from Heidemann et al. (2021), evaluated *with* and *without* EV-assistance (*higher is better*). On CIFAR10-ID, the ND drops by 71% without EV-assistance; on MNIST-ID, it drops by 19%. These results indicate that EE knowledge transfer significantly enhances ensemble diversity, which in turn contributes to better calibration.

In addition, we investigate the effect of weight transfer on model convergence. Figure 6 plots the loss after each train-batch at one of the EV-blocks of Resnet-8 on CIFAR10. As seen, model convergence is not affected because the weight transfer mechanism is designed such that the loss at EV-exit is always computed with respect to the copied weights (see Section 4).

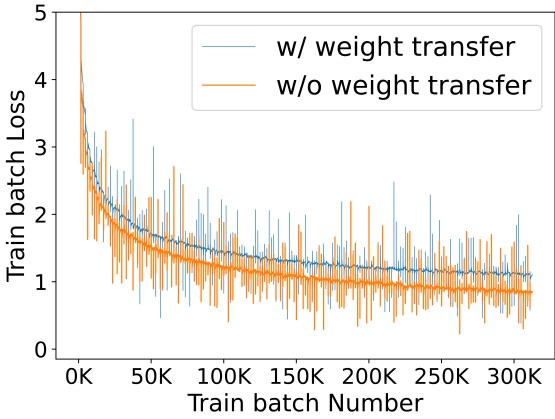

Figure 6: Batch-loss at one of the EV-exits of Resnet-8 on CIFAR10.

In essence, EV-assistance uses the EE-weights to extract and reemphasize the EE features at the final exit(s).

## B.4. Uncertainty quantification vs Ensemble size

We conduct an ablation study to investigate the effect of number of early-exits on uncertainty estimation quality. The ensemble size $|\mathcal{K}|$ is an hyperparameter that depends on the computation/resource budget, and it is bounded above by the depth of the base network. We vary the ensemble size and investigate its effect on calibration quality in MobilenetV2. Figure 7 shows the effect on accuracy and NLL in MobilnetV2 for $|\mathcal{K}|$ ranging from 2-10. The red line shows the accuracy

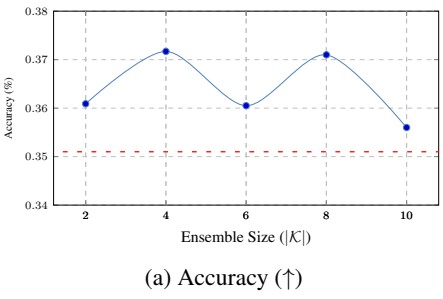

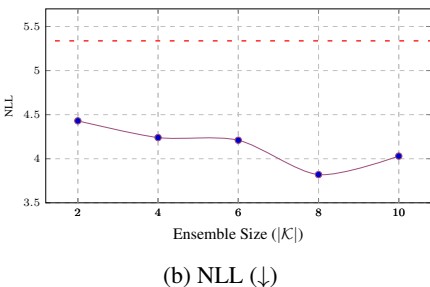

(a) Accuracy (↑)  (b) NLL (↓)

Figure 7: Effect of Ensemble size on Accuracy and NLL in MobilenetV2 on TinyImagenet. Red dotted horizontal line indicates accuracy and NLL of BASE in respective plots.

and NLL of base network in respective plots. As shown, the accuracy does not always improve with ensemble size. This demonstrates that there is a limit to the improvement in accuracy due to early-exit ensembling. On the other hand, NLL steadily improves as ensemble size increases with the least NLL obtained for $|\mathcal{K}| = 8$. For $|\mathcal{K}| = 10$, the NLL rises slightly. Our investigations revealed that since the base network, early-exits and the early-view (EV) exits are part of the same network architecture, and are trained simultaneously, the disruption in base network's weights due to the losses of early-exits and EV-exits reaches a tipping-point after a certain ensemble size thereby, causing a drop off in performance.

## B.5. Ablations with EE-ensemble

### B.5.1. EXCLUDING FINAL EXIT FROM FINAL PREDICTION

As described in Section 4, during inference, we exclude the original output block of the base network when computing the final prediction vector for QUTE because we find it that is overconfident and harms calibration quality. However, we include the original output block's prediction vector along with that of of all early-exits for computing the final prediction vector for EE-ensemble, consistent with the original paper Qendro et al. (2021). We conduct an ablation study on CIFAR10 using Resnet-8, where like QUTE, we exclude the original output block's prediction vector when computing the final prediction vector. Table 8 reports the calibration metrics on CIFAR10-ID for two configurations: 1) including original output block in computing final prediction vector and 2) excluding original output block in computing final prediction vector.

| | *Including final-exit* | | | *Excluding final-exit* | | |
|---|---|---|---|---|---|---|
| **Resnet-8** | **F1 (↑)** | **BS (↓)** | **NLL (↓)** | **F1 (↑)** | **BS (↓)** | **NLL (↓)** |
| **CIFAR10** | | | | | | |
| - EE-ensemble | 0.854 | 0.021 | 0.446 | 0.818 | 0.0026 | 0.561 |

Table 8: Calibration Metrics of EE-ensemble for Resnet-8 on CIFAR10-ID computed *including* original output block and *excluding* original output block from computation of final prediction vector.

As seen, the removal of original output block negatively impacts EE-ensemble leading to poor accuracy and calibration. Therefore, we include the original output block in computing the final prediction vector in our main results for EE-ensemble.

### B.5.2. USING QUTE'S OUTPUT BLOCK ARCHITECTURE FOR EARLY-EXITS IN EE-ENSEMBLE

Qendro et al. (2021) adds *additional fully-connected layers* at the early-exit for EE-ensemble to match the learning capacities of all exits, unlike QUTE, which uses a single depthwise CONV layer for its output blocks. However, this leads to a

large overhead (Section 6.1). In this ablation study, we show that the additional layers at the early-exit are necessary for EE-ensemble to achieve good accuracy and calibration. We replace the resource-hungry early-exits of EE-ensemble with the architecture of QUTE's output blocks.

| Resnet-8 | In-distribution | | | Corrupted-in-distribution | | |
|---|---|---|---|---|---|---|
| | F1 (↑) | BS (↓) | NLL (↓) | F1 (↑) | BS (↓) | NLL (↓) |
| **CIFAR10** | | | | | | |
| - EE-ensemble *w/ additional layers* | **0.85** | **0.021** | **0.446** | **0.64** | **0.0049** | **1.256** |
| - EE-ensemble *w/o additional layers* | 0.78 | 0.031 | 0.661 | 0.57 | 0.0058 | 1.459 |

Table 9: Calibration Metrics for Resnet-8 on CIFAR10-ID computed for EE-ensemble *with additional FC layers* and *without additional FC layers*

Table 9 reports the calibration metrics for two configurations: 1) EE-ensemble *with additional FC layers* at the early-exit and 2) EE-ensemble with QUTE's lightweight output block architecture at the early-exits. As seen, the configuration with extra learning layers clearly outperforms the one without on both ID and CID in terms of accuracy and calibration. The early-exits placed at different depths work with much less information compared to the final exit. As a result, the accuracy and calibration invariably degrade if the learning capacities of all early-exits do not match that of the final exit.

### B.6. QUTE with deeper models

To demonstrate the scalability of our proposed method, we apply QUTE to a traditional Resnet50 (He et al., 2016), a much larger model with higher learning capacity. We train Resnet50 on TinyImagenet for 50 epochs with a batch size of 128. The rest of the training methodology is the same as described in Section 4. We evaluate its accuracy-drop detection capability on TinyImagenet-corrupted. Figure 8 plots the average AUPRC over all corruptions for five severity levels.

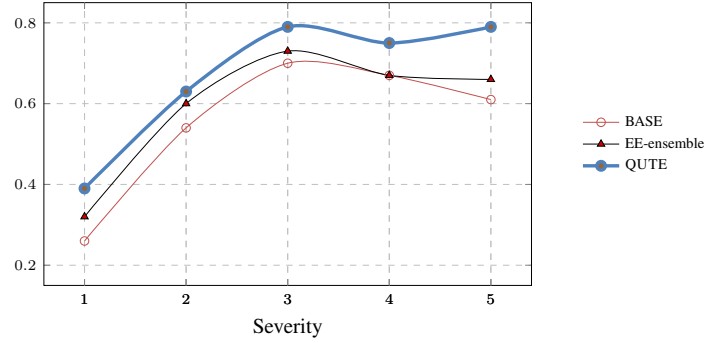

Figure 8: Accuracy-drop detection results (AUPRC) for Resnet50 on TinyImagenet-C.

As seen, QUTE comfortably outperforms EE-ensemble and BASE on accuracy-drop detection, especially showing a remarkable 19% improvement at the highest severity level compared to EE-ensemble. This illustrates QUTE's superior accuracy-drop detection capabilities across datasets/models of varying complexities, providing a highly-effective and cost-efficient accuracy monitoring mechanism.

## C. MCU results and Further Discussion

Tables 10 and 11 reports the on-device results for Big-MCU and Small-MCU respectively. We report the accuracy, code-size and the latency of each method. We also report the peak random-access-memory (RAM) usage of all methods when deployed on Big-MCU (Table 10).

On Big-MCU, all methods fit on the device. The low memory-footprint of QUTE allows it reduce the code-size consistently across datasets, leading to lower latency and, consequently, reduced energy per prediction. In contrast, EE-ensemble has higher memory requirements due to the expensive dense layers at its early exits. Furthermore, with no parallel execution capabilities on an MCU, the latency of DEEP with two ensemble members takes roughly twice as long to execute compared to BASE. This issue could worsen significantly with a larger ensemble size.

| | SpeechCmd | | | | CIFAR10 | | | |
|---|---|---|---|---|---|---|---|---|
| Model | Accuracy | Size (KB) | Latency (ms) | RAM (KB) | Accuracy | Size (KB) | Latency (ms) | RAM (KB) |
| - BASE | 0.92 | 294.5 | 22.7 | 79.3 | 0.83 | 344.4 | 58.04 | 78.5 |
| - DEEP | 0.93 | 321.6 | 45.4 | 86.2 | 0.88 | 423.4 | 116.2 | 84.2 |
| - EE-ensemble | 0.92 | 361.5 | 51.7 | 92.9 | 0.85 | 735.5 | 65.4 | 88.2 |
| - QUTE | 0.93 | 297.8 | 24.07 | 81.7 | 0.85 | 355.3 | 59.6 | 81.1 |

Table 10: Microcontroller results on Big-MCU (STM32F767ZI)

| | SpeechCmd | | | CIFAR10 | | | MNIST | | |
|---|---|---|---|---|---|---|---|---|---|
| Model | Accuracy | Size (KB) | Latency (ms) | Accuracy | Size (KB) | Latency (ms) | Accuracy | Size (KB) | Latency (ms) |
| - BASE | 0.92 | 84.8 | 160.7 | 0.83 | 187.5 | 291.2 | 0.906 | 100.3 | 5.01 |
| - DEEP | 0.93 | 111.3 | 321.2 | | `DNF/OOM` | | 0.93 | 109.3 | 9.3 |
| - EE-ensemble | 0.92 | 157.7 | 376.3 | | `DNF/OOM` | | 0.93 | 113.5 | 23.7 |
| - QUTE | 0.93 | 93 | 173.1 | 0.85 | 201 | 298.05 | 0.92 | 108.2 | 5.8 |

Table 11: Microcontroller results on Small-MCU (STM32L432KC). `DNF/OOM` indicates *did-not-fit/out-of-memory*

We observe a similar trend on the Small-MCU, with QUTE providing the best uncertainty quantification (UQ) per unit of resource used. However, both EE-ensemble and DEEP exceed the available 256KB of read-only-memory (ROM)/Flash on Small-MCU, making them unsuitable for ultra-low-power devices. In such scenarios, QUTE emerges as the only practical solution for effective UQ. Furthermore, we observe an interesting trend on the MNIST dataset. When using a 4-convolution layer model (4-layer CONV), the code sizes of all methods are quite similar, even exceeding that of the SpeechCmd BASE model, which employs a 4-layer depthwise-separable convolutional model (4-layer DSCNN) with 20,000 more parameters. A deeper analysis of the MAP files, which detail memory allocation, reveals that MNIST uses standard convolutions instead of separable convolutions. Therefore, the memory required for code is significantly higher for the 4-layer CONV model compared to the 4-layer DSCNN. In fact, the memory required for operators and code in the 4-layer CONV model is four times greater than the memory needed to store its weights.

In summary, these results highlight the efficiency of QUTE. While other methods may struggle with memory limitations, QUTE manages to deliver effective UQ without compromising on resource utilization. Its design allows for optimal performance even in constrained environments, making it a compelling choice for applications where both accuracy and efficiency are of paramount importance.

### C.1. QUTE from a system's perspective

The QUTE architecture modifies the computational structure of the model during both training and inference. During training, the addition of early-exits and extra classification heads increases the computational and memory demands. However, since we remove the early-exits during inference/deployment, retaining only the ligtweight classification heads (made up of a single depthwise-CONV layer), it significantly reduces the computational and memory burden on the low-power MCUs in deployment. The removal of early-exits also alleviates potential memory bottlenecks, particularly on devices with limited RAM.

Moreover, because all lightweight classification heads in QUTE share the output of the final CONV block as their input, the increase in peak RAM usage is minimal compared to other baselines, as shown in Table 10. For example, QUTE exhibits just a 7% increase in peak RAM usage, which is lower than the 12% average increase observed in the EE-ensemble approach because EE-ensemble requires storing of intermediate feature maps in RAM during early-exit computation. Additionally, QUTE requires only a single forward pass to replicate the behavior of an ensemble, making it far more compute- and memory-efficient than traditional ensemble methods.

Importantly, the QUTE architecture eliminates variability in prediction timing, unlike some prior methods such as cascaded deep ensembles (Xia & Bouganis, 2023). This predictability is crucial for tinyML systems, which often serve as the first component in larger ML pipelines interacting with the real world. Consistent output timing simplifies coordination with downstream tasks and ensures compatibility with systems requiring fixed output intervals.

### C.2. Applicability to Transformers and Language models

Transformer-based language and vision models have revolutionized natural language processing and computer vision, and there have been efforts to deploy them on-device in edge scenarios (Liu et al., 2024). However, the compute and memory

limitations of resource-constrained edge devices still present significant challenges for deployment in the real-world (Zheng et al., 2025). Moreover, uncertainty quantification in transformer architectures is also essential when deployed at the edge but lacks mainstream attention (Pei et al., 2022). The conventional methods for uncertainty quantification (UQ) such as MCD (Gal & Ghahramani, 2016) and ensembles (Lakshminarayanan et al., 2017) are still applicable to transformer architectures. In such scenarios, it is imperative to reduce the overhead of UQ mechanism for transformer models when deployed on tinyML or edge devices, especially when the base transformer network already incurs a high resource overhead. The QUTE architecture and the EV-assistance method can be adapted and tailored for transformer architectures. For example, before transferring knowledge from the shallow layers to the deeper layers, an auxiliary model could be added to act as an intermediary for knowledge transfer. This could potentially mitigate the structural and semantic differences between the early-exit/shallow layers and the final layers. We leave this exploration as part of the future work.

## D. Evaluation metrics

We evaluate uncertainty using the following metrics.

**Brier Score** (lower is better): It is a proper scoring rule that measures the accuracy of predictive probabilities. Incorrect predictions with high predictive confidence are penalized less by BS. Thus, BS is less sensitive to corruptions and incorrect predictions. Therefore, NLL is a better measure of uncertainty to compare with other methods.

$$BS = \frac{1}{N} \sum_{\substack{n=1 \\ l \in \{1,2,..L\}}}^{N} (p_\Theta(y = l|x) - \mathbb{1}(y = l))^2 \qquad (4)$$

where, $\mathbb{1}$ is an identity function and $L$ is the number of classes.

**Negative log-likelihood** (lower is better): It is a proper scoring rule that measures how probable it is that the predictions obtained are from the in-distribution set. It depends on both the uncertainty (predictive confidence) and the accuracy of the predictions.

$$NLL = - \sum_{l \in \{1,2,..L\}} \mathbb{1}(y = l) \cdot log p_\Theta(y = l|x) + (1 - \mathbb{1}(y = l)) \cdot log(1 - p(y = l|x)) \qquad (5)$$

**Expected Calibration Error (ECE)** (lower is better): ECE measures the absolute difference between predicted confidence and actual accuracy across different confidence intervals. The confidence is in the range [0,1]. ECE divides the confidence range into $M$ intervals/bins of size $\frac{1}{M}$ and computes the bin accuracies and confidence of each bin before averaging them to provide the final ECE score.

$$ECE = \sum_{m=1}^{M} \frac{|B_m|}{N} \cdot |acc(B_m) - conf(B_m)| \qquad (6)$$

where, $N$ is total number of predictions, $B_m$ is the $m^{\text{th}}$ bin spanning the interval $(\frac{m-1}{M}, \frac{m}{M})$. A low ECE score is desirable, and indicates less disparity between confidence and accuracy across all intervals. However, the ECE score has several limitations and susceptibilities.

### Limitations of ECE

1) ECE divides predicted probabilities into discrete bins. However, there might be a severe imbalance of samples across the bins because neural networks tend to always predict with high confidence (Nixon et al., 2019). This causes only a few bins (concentrated towards the high confidence region) to contribute the most to ECE. In addition, ECE is also sensitive to the bin boundaries i.e., the number of bins.

2) ECE does not differentiate between specific types of miscalibration such as overconfidence/underconfidence, often providing a simplistic and (overly) optimistic view of the calibration quality. Therefore, ECE is not a proper scoring metric. A model might have a lower ECE without genuinely having good calibration.

For these reasons, we do not emphasize ECE in our main results as they are not indicative of the model's true calibration quality. However, it is worth noting that there are improved versions of the ECE metric such as adaptive calibration error (ACE) and static calibration error (SCE) (Nixon et al., 2019). We focus more on proper scoring metrics like NLL and BS in our main results.

