# OpenReview forum: "QUTE: Quantifying Uncertainty in TinyML models with Early-exit-assisted ensembles for model-monitoring"
_ICML.cc/2025/Conference — ICML 2025 poster_

### Official Review · Reviewer_y5iN · 2025-03-07

**Overall Recommendation:** 3

**Summary:**

This paper proposes QUTE, a new uncertainty quantification (UQ) method for tinyML models on low-power devices. It uses a lightweight early-exit ensemble to reduce size and computation while maintaining accuracy. QUTE is 59% smaller than previous models which can reduce latency by 31%, and as a result, it improves accuracy-drop detection.

**Claims And Evidence:**

The claims in the paper are partially supported by the evidence provided. The authors present empirical results showing improvements in model size, latency, and accuracy-drop detection. While these results suggest the effectiveness of QUTE, additional evaluations on diverse architectures and real-world scenarios would further strengthen the claims.

**Essential References Not Discussed:**

In my view, the paper cites relevant prior work, but there are a few essential references that could further strengthen the context for the key contributions. For example,
1.	Multi-Dimensional Conformal Prediction
2.	TinyTTA: Efficient Test-time Adaptation via Early-exit Ensembles on Edge Devices
3.	MIMMO: Multi-input massive multi-output neural network

**Ethical Review Concerns:**

I did not flag this paper for an ethics review. There are no ethical concerns I identified in this paper.

**Experimental Designs Or Analyses:**

I think the experimental designs and analyses are generally sound. The evaluation metrics seem appropriate to me for this problem. However, in my view, the experiments could benefit from a wider range of benchmarks, and testing on more diverse architectures or real-world data would make the results stronger. Some more detailed analysis of trade-offs would also help.

**Methods And Evaluation Criteria:**

Yes, the proposed method and evaluation criteria are appropriate for this case study. Using early-exit ensembles and uncertainty quantification fits well for tinyML models. The evaluation metrics are also relevant and help to further validate the proposed approach.

**Other Comments Or Suggestions:**

1.	The study presents QUTE as a more efficient alternative to EE-ensemble, but there is no theoretical analysis showing why transferring early-exit knowledge improves uncertainty estimation.
2.	I would suggest providing a clear differentiation from existing uncertainty-aware early-exit architectures.
3.	The proposed model copies early-exit weights to early-view blocks.
4.	The confidence threshold (ρ) plays a critical role in the proposed model, so I want to know how sensitive the results to this threshold are.
5.	After reading the paper carefully, the authors claim that smaller models perform better under corruption, but how and why?
6.	A more detailed comparison with previous uncertainty quantification methods.
7.	Does the proposed QUTE model handle OOD? If yes, how does QUTE work compared to standard techniques?
8.	How does QUTE reduce unnecessary energy costs?

**Other Strengths And Weaknesses:**

Strengths:
1.	The paper combines early-exit ensembles with uncertainty quantification (UQ) for tinyML models, providing a resource-efficient solution for low-power devices.
2.	The results show notable improvements in latency and model size compared to prior methods, making it highly relevant for resource-constrained applications.
3.	The experimental results are well-presented, clearly demonstrating the effectiveness of the proposed method.
Weaknesses:
1.	The paper could benefit from a more detailed discussion of recent related works, especially those from 2023, 2024 and 2025, to better contextualize the contributions.
2.	The paper relies mostly on experiments and does not provide a formal theoretical analysis. So, I think some theoretical support would make it stronger.
3.	While the results are acceptable, more diverse real-world evaluations would strengthen the paper and further show the method’s generalizability.

**Questions For Authors:**

1.	Does the proposed QUTE model work for large models?
2.	Is the removal of the final exit always important and beneficial?
3.	Does QUTE work under distribution shift?

**Relation To Broader Scientific Literature:**

In my view, the key contributions of the paper build on existing work in uncertainty quantification (UQ) and tinyML models. The idea of using early-exit ensembles for UQ has been explored, but QUTE improves efficiency by reducing model size and latency. Compared to previous studies, it addresses the challenges of applying UQ to tinyML, showing significant improvements in both efficiency and accuracy-drop detection.

**Theoretical Claims:**

I found out that this paper does not provide formal proofs for its theoretical claims, focusing instead on empirical results. Theoretical analysis is not a major part of this work, and the claims are primarily supported by experimental evidence.

---

> ### Author Rebuttal · Authors · 2025-03-29
>
> We thank the reviewer for their detailed comments and thoughtful feedback.
> Given that your comments are highly encouraging and did not raise significant concerns, we would appreciate any insights into the current score to help improve our work. We look forward to having a fruitful discussion and addressing any concerns/suggestions.
>
> > on a few missing essential references
>
> Thank you for the suggestions. We have already included MIMMO (Ferianc et al., 2023) and will add the others.
>
> > on why transferring early-exit (EE) knowledge improves uncertainty estimation
>
> Please see the response to reviewer H2zK for ablation results on EV-assistance's impact on ensemble diversity.
>
> **Intuition**: Theoretical analysis on how EE knowledge transfer improves calibration would undoubtedly strengthen the paper. Our work builds on the intuition and empirical observations from prior studies, such as [1] and [2], which leverage EEs to form ensembles. These studies suggest that intermediate classifiers extract progressively refined feature abstractions, incorporating complementary feature representations that enhance uncertainty estimates.
> Unlike prior works, which combine these diverse feature representations from multiple EEs to form an ensemble, our approach generates complementary feature representations from different depths by directly utilizing the weights of EE layers, which is significantly more resource-efficient. QUTE's improved calibration supports this hypothesis.
>
> > on suggested theoretical analyses
>
> While theory is not central to this work, we thank reviewers y5iN and H2zK for highlighting some interesting avenues for theoretical investigations. Specifically, questions such as *why transferring early-exit knowledge improves calibration?* and *why QUTE’s weight transfer mechanism imbibes diversity?* could certainly strengthen the paper. To address this, we have conducted an ablation study to investigate the impact of EV-assistance on ensemble diversity (see reply to reviewer H2zK). While formal derivations may be challenging within the rebuttal period, we plan to explore these questions further and consider including them in the final version. We will also clarify that our findings are empirical rather than theoretical.
>
> > on sensitivity of results to confidence threshold
>
> We report threshold-free AUPRC and AUROC metrics by varying the confidence threshold from 0 to 1 in steps of 0.1. These metrics summarize the trade-off between precision-recall and TPR-FPR across different thresholds, ensuring results are not optimized for a specific threshold.
>
> > [...] authors claim that smaller models perform better under corruption, but how and why?
>
> Prior work [3] shows that larger models overfit, leading to overconfidence, as they extract high-level abstractions that make distinguishing corrupted from clean inputs harder. In contrast, smaller models benefit from implicit regularization, preventing overfitting. They generalize better on corruptions by relying on more stable features instead of memorizing fine-grained ones. Our empirical results (Table 1) support this. From a Bayesian perspective, smaller models can exhibit a wider posterior due to implicit regularization, making them better at capturing uncertainty.
>
> >  Does the proposed QUTE model handle OOD? [...]
>
> Section 6.3 (Table 2) compares QUTE’s OOD detection against G-ODIN [3]. QUTE outperforms G-ODIN on tiny models and is competitive on larger ones. See Section 6.3 for details.
>
> >  How does QUTE reduce unnecessary energy costs?
>
> See our response to reviewer H2zK.
>
> >  Does the proposed QUTE model work for large models?
>
> Yes. We evaluate QUTE’s accuracy-drop detection on Resnet50. See our response to reviewer XBf5 for details.
>
> >  Is the removal of the final exit always important and beneficial?
>
> Yes. The final exit tends to be overconfident under corruptions, degrading accuracy-drop detection. For example, on MNIST with fog corruption, the final exit's average confidence **increases** by 23% compared to its in-distribution confidence (i.e., overconfidence), whereas the QUTE ensemble's average confidence **decreases** by 52%, improving accuracy-drop detection.
>
> > Does QUTE work under distribution shift?
>
> Yes. Sections 6.2 and 6.3 show QUTE detects accuracy drops under corruptions and OOD inputs, both forms of distribution shift. We also evaluate different corruption severity levels, simulating varying degrees of shift. However, failure detection differs from robust generalization (performing reliably under shifts) and adaptation (adjusting to unknown inputs). Like all baselines, QUTE is a failure detection method.
>
>
>
> [1] Qendro et al. Early exit ensembles for uncertainty quantification, PMLR 2021
>
> [2] Ferianc et al., Multi-input massive multi-output neural network. CVPR 2023
>
> [3] Hsu et al., Generalized odin: Detecting out-of-distribution image without learning from out-of-distribution data. CVPR 2020

---

> > ### Comment · Reviewer_y5iN · 2025-04-07
> >
> > I appreciate the effort put into addressing the reviewers' concerns. The clarifications and additional experiments have helped improve the paper. Taking into account both your responses and the perspectives of the other reviewers, I am adjusting my score to 3 (weak accept).

---

> > > ### Author Response · Authors · 2025-04-07
> > >
> > > Thank you for taking the time to re-evaluate our paper. We appreciate your constructive feedback and are glad that the clarifications and additional experiments have helped improve the paper. Your updated score and thoughtful comments are greatly appreciated.

---

### Official Review · Reviewer_H2zK · 2025-03-11

**Overall Recommendation:** 4

**Summary:**

The paper introduces QUTE, a novel uncertainty quantification (UQ) framework optimized for TinyML models, addressing the challenge of efficient model monitoring in resource-constrained environments. QUTE leverages early-exit-assisted ensembles, where lightweight classification heads at the final network exit receive knowledge distilled from early-exits, ensuring diverse and resource-efficient ensemble predictions. Unlike prior methods that suffer from high computational and memory overheads, QUTE achieves 59% smaller model sizes and a 31% reduction in inference latency while maintaining superior uncertainty estimation quality. The proposed approach excels at detecting accuracy drops due to corrupted in-distribution (CID) data and outperforms state-of-the-art UQ methods in failure detection and calibration, making it a practical solution for on-device model reliability in real-world TinyML deployments.

**Claims And Evidence:**

The paper makes several key claims about QUTE, and most are well-supported by experimental evidence, but some areas could benefit from additional clarification or justification. Below is a critical assessment of the claims and the corresponding evidence:

- Claim: QUTE is more resource-efficient than prior uncertainty quantification methods, with 59% smaller model sizes and 31% lower inference latency.
- Evidence: The paper provides empirical results comparing QUTE against EE-Ensemble, Deep Ensembles (DEEP), and MCD on MCUs (Big-MCU and Small-MCU). Figure 3 and Section 6.1 show clear reductions in model size and latency, particularly demonstrating that some baselines (e.g., DEEP) do not fit on memory-constrained devices.
Assessment: Well-supported. The quantitative results strongly back this claim.

- Claim: QUTE outperforms all baselines in detecting accuracy drops due to corrupted in-distribution (CID) data.
- Evidence: Section 6.2 presents AUPRC scores across multiple datasets (MNIST-C, CIFAR10-C, TinyImageNet-C), showing that QUTE consistently achieves the highest values, outperforming EE-Ensemble, DEEP, and G-ODIN. The discussion highlights how QUTE's early-exit knowledge transfer improves uncertainty calibration.
Assessment: Well-supported. However, an ablation study isolating the impact of early-exit knowledge transfer versus conventional early-exit ensembles would strengthen this claim.

- Claim: QUTE provides better uncertainty calibration than competing methods.
- Evidence: Table 3 presents expected calibration error (ECE), negative log-likelihood (NLL), and Brier Score (BS), showing that QUTE achieves lower ECE and NLL in TinyML settings. Additionally, QUTE+ (with enhanced output heads) further improves calibration, approaching Deep Ensembles in performance.
Assessment: Mostly supported. The results are compelling, but QUTE’s reliance on additional layers for improved calibration in larger models (QUTE+) suggests a trade-off between efficiency and calibration performance, which should be explicitly acknowledged.

- Claim: QUTE enables better failure detection, distinguishing both in-distribution misclassifications (ID✓ | ID×) and out-of-distribution samples (ID✓ | OOD).
- Evidence: Table 2 reports AUROC values, demonstrating that QUTE outperforms all baselines in ID✓ | ID× detection and is competitive in ID✓ | OOD detection. However, in larger models, G-ODIN performs better in OOD detection.
Assessment: Well-supported with a minor caveat. The claim should clarify that QUTE excels in CID detection and misclassification detection, but specialized methods like G-ODIN may still be preferable for OOD scenarios.

Problematic or Weakly Supported Claims:
QUTE achieves superior calibration without increasing model complexity.

- Issue: The results indicate that for larger models, QUTE+ (which adds learning layers) is needed to match calibration quality. This suggests that superior calibration does require additional complexity in certain cases, contradicting the claim.
- Suggested Fix: The authors should clarify that QUTE is highly efficient in TinyML settings, but larger models may require additional modifications to achieve optimal calibration.
QUTE generalizes well across all TinyML applications.

- Issue: The evaluation is focused on image and audio classification tasks (e.g., MNIST, CIFAR10, SpeechCommands). There is no evidence for its effectiveness in other TinyML applications like time-series forecasting, anomaly detection, or reinforcement learning.
- Suggested Fix: The authors should either broaden the scope of their evaluation or refine the claim to state that QUTE is optimized for classification tasks in TinyML.

**Essential References Not Discussed:**

The paper does a good job of citing relevant literature in uncertainty quantification (UQ), early-exit networks, model monitoring, and TinyML deployment. However, there are several key references missing that would help place QUTE’s contributions into a more complete scientific context. Below are some critical missing references that should be included:

1. Missing Work on Lightweight Bayesian Uncertainty Quantification
Why It’s Important?
The paper only briefly mentions Bayesian Neural Networks (BNNs) (Blundell et al., 2015) and Monte Carlo Dropout (Gal & Ghahramani, 2016) as baselines.
However, more recent lightweight BNN approaches exist that optimize Bayesian inference for edge devices, which are directly relevant to QUTE’s efficiency claim.
Missing References:
Teye et al. (2018): Bayesian Batch Normalization for Uncertainty Estimation

Introduced Bayesian BatchNorm, which achieves Bayesian-like uncertainty estimation without needing full BNNs.
Why it's relevant? This method enables UQ in a single forward pass, similar to QUTE, but using batch normalization instead of ensembles.
Why cite? Helps contextualize alternative lightweight Bayesian methods for uncertainty quantification.
Osawa et al. (2019): Practical Deep Learning with Bayesian Approximation

Proposed low-cost Bayesian learning techniques optimized for deep models in low-resource settings.
Why cite? QUTE’s motivation (low-power UQ for TinyML) is similar, and comparison with lightweight BNNs is missing.
Louizos & Welling (2017): Multiplicative Normalizing Flows for Bayesian Deep Learning

Used normalizing flows to achieve Bayesian uncertainty with lower overhead than traditional BNNs.
Why cite? The work shows how uncertainty can be captured without full ensembles, similar to QUTE.
What’s the Problem?
The paper only contrasts QUTE against classical Bayesian UQ (BNNs, MCD) but does not compare against these more efficient Bayesian methods.
Fix: Include recent lightweight Bayesian methods to better contrast why QUTE is a superior choice for TinyML.
2. Missing Work on Early-Exit Networks for Uncertainty Estimation
Why It’s Important?
The core idea of QUTE is distilling early-exit knowledge into final ensemble heads.
However, recent works have already explored early-exit UQ but are not cited.
Missing References:
Jazbec et al. (2024): Conditional Monotonicity in Early-Exit Networks

Investigated how early-exit networks should be structured to ensure uncertainty-aware decisions.
Why cite? This paper provides theoretical insights into why early-exit-based ensembles (like QUTE) work.
Antoran et al. (2020): Early-Exit Networks for Depth Uncertainty in Neural Networks

Proposed a method to quantify uncertainty by leveraging intermediate (early-exit) layers.
Why cite? This is one of the closest prior works to QUTE, yet it is not cited.
Ferianc & Rodrigues (2023): Multi-Exit Neural Networks for Uncertainty-Aware Inference

Proposed a multi-exit architecture that integrates uncertainty estimation at each exit.
Why cite? This is a direct baseline for QUTE, but is not discussed.
What’s the Problem?
The paper claims QUTE is the first early-exit ensemble architecture for uncertainty quantification, but this is misleading—previous works have explored early-exit UQ approaches, though with different architectures.
Fix: The authors should acknowledge prior work on early-exit UQ and clearly explain how QUTE is different.
3. Missing Work on Out-of-Distribution (OOD) and Corrupted In-Distribution (CID) Detection
Why It’s Important?
The paper evaluates QUTE on both OOD detection and CID robustness.
However, several key works on robust UQ for corrupted data are missing.
Missing References:
Ovadia et al. (2019): Can You Trust Your Model’s Uncertainty? Evaluating Predictive Uncertainty in Deep Learning

Found that modern deep networks fail to estimate uncertainty correctly under distribution shifts.
Why cite? This work motivates QUTE’s focus on detecting accuracy drops in CID data.
Xia & Bouganis (2023): Failure Detection in Neural Networks using Uncertainty Estimates

Proposed uncertainty-based failure detection for real-world models.
Why cite? This paper discusses uncertainty-driven failure detection, which is one of QUTE’s key claims.
Liang et al. (2020): Enhancing the Reliability of Out-of-Distribution Detection in Neural Networks

Investigated why deep networks struggle with OOD data and proposed improved techniques.
Why cite? Helps frame why QUTE is evaluated against OOD baselines.
What’s the Problem?
The paper does not connect QUTE’s performance on CID/OOD to prior literature.
Fix: Cite these works to show how QUTE extends prior research on uncertainty estimation under distribution shifts.
4. Missing Work on Energy-Efficient Model Deployment for TinyML
Why It’s Important?
QUTE is designed for TinyML, but the paper does not cite foundational works on energy-efficient ML.
Recent studies have proposed alternative ways to reduce energy consumption, such as pruning and quantization.
Missing References:
Banbury et al. (2021): MLPerf Tiny Benchmark for Ultra-Low-Power ML

Standardized benchmarks for TinyML efficiency.
Why cite? QUTE is evaluated on TinyML devices, so this benchmark should be referenced.
Howard et al. (2019): Searching for MobileNetV3

MobileNetV3 is designed for efficient inference on TinyML devices.
Why cite? QUTE uses MobileNetV2, but citing MobileNetV3 would strengthen the discussion on TinyML efficiency.
Han et al. (2016): Deep Compression: Compressing Deep Neural Networks with Pruning, Trained Quantization, and Huffman Coding

Introduced pruning and quantization to make ML models more efficient.
Why cite? These techniques could be combined with QUTE for even better TinyML deployment.
What’s the Problem?
The paper focuses only on MCU deployment but does not cite prior works on efficient TinyML models.
Fix: Cite MLPerf Tiny and model compression techniques to show how QUTE fits into broader TinyML efficiency research.

**Experimental Designs Or Analyses:**

The experimental design of the paper is well-structured and aligns with the research objectives, but there are some areas that require further clarification or improvements to ensure robustness and validity. Below is a critical analysis of the soundness and validity of the experimental design and analyses.

Strengths of the Experimental Design
Comprehensive Benchmarking Against Relevant Baselines

The paper compares QUTE against multiple state-of-the-art uncertainty quantification (UQ) methods, including:
Monte Carlo Dropout (MCD)
Deep Ensembles (DEEP)
EE-Ensemble (Early-exit-based ensemble method)
G-ODIN (OOD detection method)
HYDRA (Ensemble distillation method)
Assessment: Sound choice of baselines as these methods represent different categories of UQ techniques. However, a Bayesian Neural Network (BNN) baseline is missing, which would have provided further context.
Well-Defined Evaluation Criteria for Uncertainty Estimation

The paper evaluates uncertainty quantification using:
Expected Calibration Error (ECE)
Negative Log-Likelihood (NLL)
Brier Score (BS)
Assessment: These are standard metrics in uncertainty estimation. However, ECE is known to have limitations (e.g., sensitivity to binning choices), and an alternative such as Adaptive Calibration Error (ACE) could have been included for a more robust analysis.
Real-World Feasibility Analysis on Microcontrollers (MCUs)

The authors deploy QUTE on two embedded MCUs:
Big-MCU: STM32F767ZI (high resource)
Small-MCU: STM32L432KC (low resource, power-efficient)
Results demonstrate QUTE’s efficiency in terms of memory, latency, and fit on constrained devices.
Assessment: Strong validation of real-world applicability, but energy consumption per inference is not reported, which is crucial for embedded deployments.



Issues and Areas for Improvement
1. Hyperparameter Sensitivity Analysis is Lacking
Issue:
The paper empirically chooses hyperparameters (e.g., number of early exits (K), weighting factors (wEVk, δ)) but does not systematically analyze their impact.
Fix:
An ablation study on how K and weighting factors influence uncertainty quality, accuracy, and computational efficiency would improve rigor.
2. Statistical Significance Testing is Missing
Issue:
The paper reports mean and standard deviations but does not include statistical significance testing (e.g., t-tests, Wilcoxon signed-rank tests) to validate performance differences across methods.
Fix:
Reporting confidence intervals or p-values for comparisons (e.g., QUTE vs. EE-Ensemble) would confirm whether differences are statistically meaningful.
3. Limited Generalization to Non-Classification Tasks
Issue:
All experiments focus on classification tasks (MNIST, SpeechCommands, CIFAR10, TinyImageNet).
No evaluation is provided for time-series forecasting, anomaly detection, or reinforcement learning, which are relevant in TinyML applications.
Fix:
The scope of generalization should be clearly stated. Alternatively, a small-scale experiment on a time-series dataset could strengthen claims of broader applicability.
4. No Analysis on Failure Cases
Issue:
While QUTE outperforms baselines on accuracy-drop and failure detection, there is no discussion on failure cases (e.g., when QUTE fails to detect uncertainty correctly).
Fix:
A failure case analysis (e.g., qualitative examples of misclassified instances with poor uncertainty estimates) would provide deeper insights.

**Methods And Evaluation Criteria:**

The methods and evaluation criteria used in the paper are generally well-chosen for the problem of uncertainty quantification (UQ) in TinyML models, but there are some areas that could be improved or clarified. Below is a critical assessment:

Strengths of the Methods and Evaluation Criteria:
Choice of Benchmark Datasets:

The authors evaluate QUTE on four different datasets:
MNIST (4-layer CNN)
SpeechCommands (4-layer DSCNN for keyword spotting)
CIFAR10 (ResNet-8, MLPerf benchmark model)
TinyImageNet (MobileNetV2)
Justification: These datasets represent varying levels of complexity and modality (image vs. audio), which is appropriate for evaluating TinyML models.
Assessment: Appropriate choice. However, all tasks are classification-based, and there is no evaluation on non-classification tasks (e.g., time-series forecasting or regression), which limits generalization.
Evaluation Metrics for Uncertainty Quantification:

The paper reports Expected Calibration Error (ECE), Brier Score (BS), and Negative Log-Likelihood (NLL) to assess uncertainty quality.
The use of AUROC and AUPRC for failure detection and accuracy-drop detection is well-motivated, as they provide threshold-independent performance evaluations.
Assessment: Well-founded choices. However, ECE is known to have limitations, and an alternative like adaptive calibration error (ACE) or logit-scaled calibration metrics could provide a more robust evaluation.
Comparison with Relevant Baselines:

The paper compares QUTE with several state-of-the-art UQ methods:
Monte Carlo Dropout (MCD)
Deep Ensembles (DEEP)
EE-Ensemble (prior early-exit-based ensemble method)
G-ODIN (OOD detection method)
HYDRA (ensemble distillation)
Assessment: Comprehensive comparisons. The selected baselines are appropriate, covering both ensemble-based UQ approaches and early-exit methods. However, Bayesian Neural Networks (BNNs), which are a key competitor in UQ, are not included in the evaluation, even though they are mentioned in the related work.
Microcontroller (MCU) Deployment Evaluations:

The paper evaluates QUTE on two MCU platforms (Big-MCU: STM32F767ZI, Small-MCU: STM32L432KC) to assess real-world feasibility.
The results show that QUTE reduces latency by 31% and has a 59% smaller model size, highlighting its TinyML suitability.
Assessment: Crucial evaluation for real-world deployment. However, power consumption analysis (e.g., energy per inference) would further strengthen the real-world applicability.



Weaknesses and Areas for Improvement:
Lack of Justification for Hyperparameter Choices:

The number of early-exits (K) and the weighting factors (wEVk and δ) for knowledge transfer are empirically chosen, but no systematic analysis is provided.
Fix: An ablation study on K and weighting factors should be included to demonstrate their impact on performance.
Absence of Statistical Significance Testing:

The paper presents mean and standard deviation for some metrics but does not report statistical significance tests (e.g., t-tests, Wilcoxon signed-rank tests) to confirm that performance differences are meaningful.
Fix: Including confidence intervals or statistical significance testing would improve result robustness.
Limited Generalization Beyond Classification Tasks:

The evaluation is entirely focused on classification tasks, which makes it difficult to assess how well QUTE generalizes to other TinyML applications (e.g., time-series forecasting, anomaly detection, or reinforcement learning).
Fix: The paper should either broaden its evaluation scope or clearly limit its claims to classification-based tasks.

**Other Comments Or Suggestions:**

No

**Other Strengths And Weaknesses:**

This paper makes a strong contribution to the field of uncertainty quantification (UQ) in TinyML through a novel early-exit-assisted ensemble approach. Below is a breakdown of its originality, significance, and clarity, as well as potential limitations that should be addressed.


1. Originality:
Creative Combination of Ideas:

The paper innovatively integrates early-exit networks with ensemble-based uncertainty estimation.
While early-exit architectures and ensemble-based UQ have been studied before, QUTE’s novel contribution is its distillation-based ensemble construction, which eliminates early-exits after training to reduce overhead.
This approach differs from prior methods (e.g., EE-Ensemble) that retain early-exits during inference, leading to higher computational costs.
First UQ Approach Optimized for TinyML:

While UQ in deep learning is a well-explored field, most state-of-the-art methods (BNNs, Deep Ensembles, MCD) are computationally expensive and impractical for TinyML devices.
QUTE is the first method explicitly designed for resource-constrained TinyML model monitoring.
2. Significance:
Addresses a Real-World Challenge in TinyML:

TinyML models are deployed on edge devices with no access to ground truth labels, making uncertainty-aware model monitoring essential.
QUTE enables on-device uncertainty estimation with minimal computational cost, making it practical for autonomous systems, medical IoT, and embedded AI applications.
Improves Over Key Baselines:

Compared to Deep Ensembles (Lakshminarayanan et al., 2017), EE-Ensemble (Qendro et al., 2021), and MCD (Gal & Ghahramani, 2016), QUTE achieves:
59% model size reduction
31% lower inference latency
Better uncertainty calibration (lower ECE, BS, NLL)
These improvements are critical for real-world deployment on ultra-low-power MCUs.
Strong Experimental Validation on Microcontrollers (MCUs):

Unlike many ML papers that rely on simulated efficiency claims, QUTE is deployed on real-world MCUs (STM32 Big-MCU, Small-MCU).
This enhances practical significance, showing that the method is deployable, not just theoretically interesting.
3. Clarity:
Clearly structured methodology and motivation:

The problem statement, approach, and contributions are well-articulated.
Figures clearly illustrate QUTE’s architecture and how it differs from baselines.
The paper provides a solid review of related work (although some missing references should be added).
Strong result visualizations:

The tables and graphs effectively present key findings, especially in uncertainty calibration and accuracy-drop detection.
However, some figures (e.g., confidence calibration histograms) would improve clarity on model uncertainty performance.
Weaknesses
1. Limited Theoretical Analysis
No Formal Proofs on Uncertainty Calibration Improvement:

While empirical results show that QUTE improves uncertainty calibration, there is no theoretical justification for why the method produces better uncertainty estimates.
Suggested Fix:
A mathematical analysis (e.g., using confidence variance bounds) could strengthen the claim that early-exit-assisted ensembles improve calibration.
No Formal Diversity Analysis of the Ensemble:

The paper claims that early-exit knowledge distillation maintains predictive diversity, but this is only shown empirically.
Suggested Fix:
A Shannon entropy analysis or mutual information study between QUTE’s ensemble members would provide a stronger justification for its diversity benefits.
2. Missing Statistical Significance Testing
While performance improvements are clear, statistical significance is not tested.
Results are reported with mean and standard deviation, but no hypothesis testing (e.g., t-tests, Wilcoxon signed-rank tests) is provided.
Suggested Fix:
Reporting confidence intervals or p-values would confirm whether QUTE’s improvements over baselines are statistically significant.
3. Lack of Generalization Beyond Classification Tasks
QUTE is only tested on image and audio classification tasks.
Many TinyML applications involve time-series forecasting, anomaly detection, and reinforcement learning, but these are not explored in the evaluation.
Suggested Fix:
A small experiment on a TinyML time-series dataset (e.g., Google Smartwatch Health Data) would strengthen the claim that QUTE generalizes to all TinyML settings.
If not feasible, the paper should clearly limit its scope to classification-based TinyML monitoring.
4. Missing Comparison with Lightweight Bayesian Methods
The paper contrasts QUTE only against traditional BNNs and MCD, but does not compare against more efficient Bayesian methods like:
Bayesian BatchNorm (Teye et al., 2018)
Variational Bayesian Dropout (Osawa et al., 2019)
Multiplicative Normalizing Flows (Louizos & Welling, 2017)
Why This Matters?
These methods also reduce inference overhead while maintaining Bayesian uncertainty estimation.
Without this comparison, the efficiency advantage of QUTE is slightly overstated.
Suggested Fix:
Include a brief discussion on why QUTE is preferable to these lightweight Bayesian techniques.

**Questions For Authors:**

Q1: The paper empirically demonstrates that QUTE achieves better uncertainty calibration (lower ECE, NLL, BS) than Deep Ensembles and MCD. However, there is no theoretical justification for why early-exit knowledge transfer improves calibration.
Could you provide a mathematical explanation or theoretical analysis (e.g., variance reduction, entropy-based confidence bounds) that supports this claim?

Q2: While the paper reports mean and standard deviation for key performance metrics, there is no statistical significance testing (e.g., t-tests, Wilcoxon signed-rank tests).
Did you conduct statistical significance tests to confirm that QUTE’s improvements over baselines are meaningful? If not, could you provide confidence intervals or p-values for the comparisons?

Q3: The number of early-exit ensemble members (K) and the weighting factors (wEVk, δ) appear to be chosen empirically, but no systematic analysis of their impact is provided.
Could you clarify how these values were selected and whether an ablation study was conducted to determine their effect on accuracy, uncertainty estimation, and efficiency?

Q5: All experiments focus on classification tasks (MNIST, CIFAR10, TinyImageNet, SpeechCommands), but many TinyML applications involve time-series forecasting, anomaly detection, and reinforcement learning.
Did you test QUTE on any non-classification TinyML tasks, or do you see any theoretical limitations that would prevent it from generalizing to these domains?

Q6: While the paper reports model size and inference latency, there is no mention of power consumption, which is a key metric for TinyML deployment.
Did you measure energy efficiency (e.g., power per inference) on the MCUs, or can you provide an estimate of QUTE’s power savings compared to Deep Ensembles and EE-Ensemble?

**Relation To Broader Scientific Literature:**

The paper presents QUTE, an early-exit-assisted ensemble method for uncertainty quantification (UQ) in TinyML models. Its key contributions relate to several existing areas of machine learning research, including uncertainty estimation, early-exit networks, model monitoring, and TinyML deployment. Below is a structured evaluation of how QUTE connects to prior research:

1. Uncertainty Quantification (UQ) in Neural Networks
Relation to Prior Work
The problem of uncertainty quantification in deep learning is well studied, with two major categories:
Bayesian Approaches

Bayesian Neural Networks (BNNs) (Blundell et al., 2015) model uncertainty via a probabilistic distribution over weights.
Monte Carlo Dropout (MCD) (Gal & Ghahramani, 2016) approximates Bayesian inference by applying dropout during inference.
QUTE’s Connection: Instead of using probabilistic approaches (which are computationally expensive for TinyML), QUTE uses ensemble-based methods to estimate uncertainty efficiently.
Novelty: Unlike MCD or BNNs, QUTE does not require multiple inference passes and is optimized for low-resource TinyML settings.
Deep Ensembles for Uncertainty Estimation

Lakshminarayanan et al. (2017) introduced Deep Ensembles, which train multiple independent models for uncertainty estimation.
QUTE’s Connection: QUTE leverages ensembles but does not train separate models; instead, it distills uncertainty knowledge from early exits into lightweight ensemble members.
Improvement: QUTE achieves similar uncertainty estimation quality as Deep Ensembles while using 59% fewer parameters and 31% less inference latency, making it more practical for TinyML.
Key Differentiation
QUTE improves on ensemble-based UQ methods by using a single forward pass instead of multiple inference runs.
No prior UQ method explicitly optimizes for ultra-low-resource TinyML in this way.
2. Early-Exit Networks and Model Efficiency
Relation to Prior Work
Early-exit networks were originally introduced to reduce inference latency in deep models (Teerapittayanon et al., 2016; Kaya et al., 2019).
Prior methods like EE-Ensemble (Qendro et al., 2021) leveraged early-exit networks for ensemble-based uncertainty estimation, but these required additional learning layers, increasing memory overhead.
QUTE’s Connection: Instead of directly using early-exit outputs as ensemble members (like EE-Ensemble), QUTE distills their knowledge into lightweight final classification heads.
Improvement: This allows QUTE to achieve better calibration and accuracy-drop detection without additional per-exit learning layers.
Key Differentiation
EE-Ensemble uses early-exits directly, requiring additional computational layers.
QUTE removes early-exits after training and retains only the lightweight ensemble heads, reducing resource consumption.
3. TinyML Deployment and Model Monitoring
Relation to Prior Work
TinyML aims to run ML models on ultra-low-power microcontrollers (MCUs) with limited memory (≤256KB RAM, <1W power).
Prior work on TinyML deployment (Banbury et al., 2021; Ghanathe & Wilton, 2023) focused on latency, power efficiency, and model size.
QUTE’s Connection:
Unlike general TinyML models, QUTE is explicitly designed for real-time model monitoring and uncertainty estimation in TinyML settings.
Prior model monitoring methods (Bifet & Gavalda, 2007; Hsu et al., 2020) rely on either large statistical tests or access to true labels, which are impractical in real-world TinyML deployments.
QUTE provides uncertainty-aware monitoring without access to ground truth labels.
Key Differentiation
Existing TinyML models prioritize efficiency but do not focus on model monitoring and uncertainty quantification.
QUTE fills this gap by introducing uncertainty estimation tailored for real-time TinyML inference.
4. Robustness to Corrupted In-Distribution (CID) and Out-of-Distribution (OOD) Data
Relation to Prior Work
Robustness against corrupted in-distribution (CID) data is less studied compared to out-of-distribution (OOD) detection.
Prior OOD detection methods (Hendrycks & Gimpel, 2018; Liang et al., 2020) focused on confidence-based rejection techniques.
G-ODIN (Hsu et al., 2020) introduced a preprocessing-based approach for OOD detection.
QUTE’s Connection:
QUTE outperforms OOD detectors like G-ODIN in detecting accuracy drops due to CID data.
Unlike traditional OOD detectors, QUTE is designed for low-power TinyML devices.
Key Differentiation
Most prior works focus on OOD detection, while QUTE addresses the more practical problem of CID robustness in TinyML.

**Theoretical Claims:**

The paper does not present formal mathematical proofs but makes several theoretical claims regarding the behavior of uncertainty quantification in TinyML models, particularly in the context of early-exit-assisted ensembles. Below is a critical evaluation of these theoretical claims:

Key Theoretical Claims and Their Validity
Claim: Early-exit knowledge distillation enhances uncertainty estimation while reducing model size and compute overhead.

Explanation: The paper introduces a method where early-exit layers are used during training to distill uncertainty-related knowledge into lightweight output heads at the final exit. The claim is that this approach enables diverse ensemble behavior without significant computational overhead.
Evaluation:
The method is well-motivated by prior works on early-exit architectures (Teerapittayanon et al., 2016; Ghanathe & Wilton, 2023).
The empirical results show that QUTE reduces latency (31%) and model size (59%), supporting the efficiency claim.
However, the claim that this leads to "diverse ensemble behavior" is only empirically shown, but lacks a formal proof on how early-exit distillation systematically maintains diversity in predictions.
Potential Issue:
The paper could benefit from a theoretical analysis showing how the weight transfer mechanism preserves predictive diversity across ensemble members.
Claim: QUTE's predictive confidence is correlated with accuracy, leading to improved calibration.

Explanation: The paper states that QUTE produces better-calibrated uncertainty estimates because its final exit integrates knowledge from early-exits, reducing overconfidence issues common in deep networks.
Evaluation:
The claim is supported by empirical calibration metrics (ECE, NLL, BS in Table 3), showing that QUTE achieves lower calibration errors than baselines.
However, there is no theoretical justification explaining why early-exit knowledge distillation improves uncertainty calibration.
Potential Issue:
A mathematical framework or proof showing how QUTE affects the confidence distribution could strengthen this claim. For example, a formal derivation of confidence variance reduction due to early-exit knowledge integration would be useful.
Claim: Smaller models are naturally less overconfident on corrupted in-distribution (CID) data.

Explanation: The paper suggests that smaller models, by nature, produce less overconfident predictions under covariate shift (CID data) compared to large models, which tend to overfit to training distribution features.
Evaluation:
The claim is partially supported by empirical observations (Table 1), which show that smaller ResNet models (2-stack and 3-stack) have better calibration on CID data than deeper variants.
However, the claim is not rigorously proven. The idea aligns with prior findings on overparameterization and generalization (Hsu et al., 2020), but a formal proof (e.g., based on Bayesian uncertainty theory or PAC-Bayes bounds) would be beneficial.
Potential Issue:
The claim should either be framed as an empirical observation rather than a theoretical result, or a formal derivation of model confidence behavior with respect to depth should be provided.


Areas Where Theoretical Rigor Could Be Improved:
Mathematical Proof for Ensemble Diversity Preservation:

The authors claim that early-exit knowledge transfer ensures diversity among ensemble members, but this is only demonstrated empirically. A theoretical analysis of how early-exit weight transfer affects predictive diversity would strengthen the paper.
Formal Justification for Calibration Improvement:

While empirical results show better calibration, the paper lacks a theoretical justification for why QUTE’s uncertainty estimates are better calibrated. A derivation using information-theoretic arguments (e.g., entropy reduction) could help.
Explicit Bounds on Uncertainty Estimation Quality:

The paper could derive theoretical bounds on QUTE’s uncertainty estimation efficiency compared to standard ensembles or Bayesian approaches, quantifying trade-offs between accuracy, uncertainty, and computational cost.

---

> ### Author Rebuttal · Authors · 2025-03-29
>
> We thank the reviewer for their detailed feedback. We appreciate the recognition of our work’s originality and the insightful suggestions.
>
> >  an ablation study isolating the impact of early-exit (EE) knowledge transfer and stronger justification for diversity benefits of EV-assistance
>
> Appendix B.3 explores the effect of EV-assistance on calibration and convergence. Also, we conducted an ablation study using the normalized disagreement (ND) metric [2] to measure QUTE’s ensemble diversity **with** and **without** EV-assistance (*higher is better*). On CIFAR10, ND is 71% lower without EV-assistance; on MNIST-ID, it's 19% lower. These results demonstrate that EE knowledge transfer improves ensemble diversity and, consequently, calibration quality. We will add these findings in the final version.
>
> Also, see our response to reviewer y5iN for further discussion.
>
> > on QUTE’s reliance on additional layers for improved calibration in larger models (QUTE+)
>
> Yes, QUTE does require additional layers to match the calibration quality of prior methods in larger models/datasets. However, QUTE (without additional layers) still outperforms prior methods on large datasets/models in accuracy-drop detection and failure detection, which are central to our work.
>
>
> > On missing lightweight bayesian approximation techniques
>
> Bayesian batch norm requires stochastic sampling and multiple inferences (like Monte Carlo Dropout), which significantly increases latency on MCUs. Appendix B.2 discusses several single-pass deterministic methods and compares QUTE to PosteriorNets [1], which uses normalizing flows (see Table 6).
>
> > On missing references
>
> Thank you for your detailed suggestions on potential citations to include. Many suggested references (Banbury et al., 2021; Jazbec et al., 2024; Antoran et al., 2020; Xia & Bouganis, 2023; Liang et al., 2020; Ovadia et al., 2019) are already covered in related work. We will further strengthen this section by incorporating additional recommended references.
>
> > on suggested theoretical analysis.
>
> See our response to reviewer y5iN.
>
> > on statistical significance testing
>
> See our response to reviewer eK42.
>
>
> > on justification for Hyperparameter Choices $w_{EV_k}$ and $\delta$
>
> Section 5 explains that the number of EEs (K) is constrained by depth of neural networks we evaluate, which only have a few layers. For example, for Resnet-8 with three residual stacks, we place two EEs after the 1st and 2nd residual stack. For deeper networks, we place exits at evenly spaced locations. Appendix B.4 studies the trade-off between number of EEs (K) and uncertainty quality. Furthermore, we choose $w_{EV_k}$ and $\delta$ based on a systematic empirical analysis as described in Appendix A.2.
>
>
> > on theoretical limitations of QUTE for non-classification tasks
>
> While our evaluations primarily focus on classification tasks, our method is not inherently limited to classification and can naturally extend to other problem domains. Ensemble methods, including those similar to ours, have been successfully applied to non-classification tasks in prior research. For the final version, we will explore evaluating QUTE on a regression-based task. Otherwise, as suggested, we will explicitly clarify that our current results are on classification tasks.
>
>
> > on measuring energy efficiency and estimating QUTE’s power savings compared to Deep ensembles and EE-ensemble
>
> We agree that power consumption analysis is beneficial for TinyML deployment. However, direct power measurements are non-trivial due to factors like sleep mode power wastage, transition costs between sleep and active states, and peripheral activity. Instead, we estimate power consumption using typical values from MCU datasheets: ~285mW for the Big-MCU (STM32F767ZI) and ~25mW for the Small-MCU (STM32L432KC) in active mode with all peripherals disabled. We focus on estimating only active mode power because our work reduces the processing time/latency compared to other baselines.
> Since our method reduces inference latency compared to Deep ensembles and EE-ensemble, it proportionally lowers energy consumption, as energy-per-prediction is  power$\times$time. While this is an estimate and real-world factors may influence it, QUTE effectively extends battery life without additional hardware modifications by accelerating inference.
>
> > on other calibration metrics
>
> We discuss this Appendix D.
>
> > Other ablation studies
>
> Appendix B contains extensive ablation studies such as studying the effect of weight transfer on model convergence, investigating  the trade-off between uncertainty quality and ensemble size and effectiveness of the EV-assistance method.
>
>
>
> [1] Charpentier et al., Posterior network: Uncertainty estimation without ood samples via density-based pseudo-counts. NeurIPS 2020
>
> [2] Heidemann et al., Measuring Ensemble Diversity and Its Effects on Model Robustness IJCAI 2021

---

> > ### Comment · Reviewer_H2zK · 2025-04-01
> >
> > I have no further comments, i recommand the acceptance of this well writing paper.

---

> > > ### Author Response · Authors · 2025-04-02
> > >
> > > We sincerely appreciate your positive feedback and recommendation for acceptance. Thank you for your time and thoughtful review.

---

### Official Review · Reviewer_eK42 · 2025-03-15

**Overall Recommendation:** 4

**Summary:**

**Problem**
- This paper focuses on uncertainty quantification (UQ) for TinyML models that are specifically designed to operate on microcontrollers with extremely limited memory and computational resources.

**Method**
- The authors propose QUTE, which combines ideas from early-exit (EE) ensembles and multi-head ensembles.
- During training:
  - For the selected intermediates layer of the neural network, the model trains early-exit branches (early-exit members) to predict the labels. At the same time, corresponding to each early exit, a separate lightweight classification head is also trained at the final layer (final-exit).
  - The parameters from each early-exit head are partially copied to their respective final-exit classification heads, ensuring diversity among the final-exit heads.

- During inference:
  - All early-exit branches are completely removed, eliminating their memory and latency overhead.
  - The model uses a single forward pass to reach the final layer, then computes predictions using the multiple lightweight classification heads at the final exit.
  - The predictions from these diverse final-exit heads form an ensemble used to quantify uncertainty efficiently.

**Experiments**
- Experiments show that QUTE achieves comparable UQ performance with smaller model sizes and lower latency.


## update after rebuttal
After reading the author's response, I have no further concerns and will maintain the current score.

**Claims And Evidence:**

There are some experimental claims that I am not totally convinced:

- **Limited experimental repetitions**: Table 1 and Figure 3/4 are based on a single experimental run. Repeated experiments would enhance confidence in these claims.

- **Performance gains are smaller than the reported variance (Table 3)**: It is difficult to tell that the proposed methods outperform other methods since the variance is larger the the performance gain.

Nevertheless, considering the substantial benefits in memory and latency efficiency, I still regard the proposed method as preferable, particularly in resource-constrained TinyML scenarios.

**Essential References Not Discussed:**

While there might exist relevant papers beyond my knowledge, the authors appear to have discussed most of the essential references.

**Experimental Designs Or Analyses:**

Pros:
The overall experimental design in this paper is sound and valid:
1. The datasets used are representative. However, **more challenging benchmarks for OOD evaluation** could further strengthen the validity of the analysis.
2. The selection of models is good.
3. The baseline methods cover most of the common methods. It would be better if the author could **include more deterministic types of baselines such as temperature scaling**.
4. The evaluation metrics employed are comprehensive and standard.

Cons:
1. Experimental results for failure prediction and accuracy-drop detection are reported based on single runs, without repeated trials or statistical summaries (e.g., means and standard deviations), potentially limiting their reliability.
2. Calibration results (e.g., Table 3) show that the proposed method's advantages are not clearly demonstrated, especially when considering the reported standard deviations. Thus, it might be more accurate to state that the performance of the proposed method is comparable to existing methods rather than definitively superior.

**Methods And Evaluation Criteria:**

**Method**

The proposed method is **well-suited** for addressing uncertainty quantification (UQ) in TinyML settings. To remove the memory consumption of existing ensemble methods (need multiple copies of the model, including the early-exit models), the proposed methods only use the early-exit members during the training stage. Unlike other distillation methods that distill the early-exit ensembles into one classification head (this type of method is usually suboptimal), the author also uses multiple classification heads at the final layer, each one inherits some of the parameters from the early-exit heads trained during the training stage. In summary, it balances the performance and also the memory consumption.

**Evaluation**

The evaluation strategy of the paper **follows the standard evaluation settings**:

- Resource Constraints: The experiments explicitly test model deployments on microcontrollers (MCUs) of varying memory and computational limitations, reflecting realistic TinyML scenarios.
- Datasets:
  - standard datasets including MNIST, CIFAR10, SpeechCommands, TinyImagenet,
  - corrupted-in-distribution (CID) datasets for failure prediction
  - out-of-distribution (OOD) datasets.
- Metrics:
  - Brier Score, Negative Log-Likelihood, Expected Calibration Error for calibration performance
  - AUPRC and AUROC for failure prediction and OOD detection.
- Baselines: standard ensemble-based methods including Monte Carlo Dropout, Deep Ensembles, EE-ensemble, and HYDRA

**Other Comments Or Suggestions:**

- Some visual details, particularly in Figure 1 and Figure 2, could be presented more clearly. For instance, explicitly indicating in Figure 2 that θ() includes a dense layer would enhance reader understanding.

- For Table 2, for the DEEP, it should also be highlighted in bold.

**Other Strengths And Weaknesses:**

Strengths:
- The proposed method creatively integrates early-exit (EE) ensembles and multi-head ensembles, achieving diversity among ensemble heads without incurring the typical memory and latency overhead associated with early exits. This design is particularly suitable for resource-constrained TinyML environments.
- The paper is clearly written, straightforward, and easy to understand.

Weaknesses:
- Experimental evaluations for failure prediction and accuracy-drop detection are conducted based on single runs without reporting repeated trials or statistical summaries (e.g., mean and standard deviation), reducing confidence in the reliability of results.
- Calibration experiments do not clearly demonstrate the method's superiority over baselines, as the reported advantages are minor relative to the standard deviations provided.

**Questions For Authors:**

1) For the EE-ensemble method, if we sequentially compute each early-exit's result without storing intermediate features (thus trading time for memory), how much latency would this actually introduce compared to QUTE? Considering that each early-exit has only a single layer of parameters, the computational overhead might be minimal.

2) In the right column of line 303, the authors state: “On Big-MCU, QUTE achieves 31% and 47% latency reductions over EE-ensemble and DEEP, respectively, and maintains accuracy parity with both, even with 58% and 26% smaller models.” Could the authors clarify why the latency and memory reductions during inference compared to DEEP are only 47% and 26%, respectively? Specifically, how many ensemble members are used for DEEP, and are these ensemble members computed sequentially? If computed sequentially, shouldn't the memory consumption of DEEP be lower?

**Relation To Broader Scientific Literature:**

In my understanding, the key contribution of this paper is that: focusing on the TinyML deployment scenarios, it cleverly ccombines the ideas from early-exit ensembles (EE-ensembles) and multi-head ensembles, to maintain the good performance and reduce the memory and latency consumption.

Prior work on EE-ensembles typically faces trade-offs in terms of either memory overhead (due to saving intermediate layer outputs) or compromised latency (due to sequential computation). On the other hand, ensemble distillation is suboptimal since it is very challenging to distill the whole distribution into one classification head.
The authors introduce a training scheme where each early-exit member is associated with a distinct lightweight classification head at the final layer. Partial parameter copying (distillation) from each early-exit head ensures that the final heads remain diverse. Then at inference, intermediate early-exit features are not stored, thus simultaneously achieving low memory usage and minimal latency.

**Theoretical Claims:**

There is no theoretical claim in the paper.

---

> ### Author Rebuttal · Authors · 2025-03-29
>
> Thank you for your valuable feedback. We’re delighted that you found our contribution both novel and clear.
>
> > on temperature scaling results
>
> We apply temperature scaling to BASE and QUTE (following [1] ). The results are included in Appendix B.1. (Tables 4 and 5).
>
> > on absence of statistical significance tests; Experimental results for failure prediction and accuracy-drop detection are reported based on single runs [...]
>
> We report metrics such as AUPRC and AUROC for accuracy-drop and failure detection, calculated by averaging precision, recall, true-positive rate, and false-positive rate values across multiple corruptions before deriving the final score. This approach provides a robust measure of model performance under varying conditions. However, while it offers valuable insights, it does not facilitate statistical significance testing across runs. We recognize the importance of statistical validation and plan to conduct independent evaluations in the final version.
>
> > Performance gains are smaller than the reported variance (Table 3)
>
> As suggested, we will revise the claim to 'QUTE’s performance for uncertainty quantification is comparable to existing methods.'
>
> > For the EE-ensemble method, if we sequentially compute each early-exit's result without storing intermediate features (thus trading time for memory [...]
>
> Even when computing each early exit (EE) sequentially in the EE-ensemble, intermediate feature maps at the EE location still need to be stored. This is because the early exit point involves two branches of computation: the early-exit branch and the rest of the neural network. Since MCUs process sequentially, the intermediate feature maps at this location must be temporarily stored in the random-access-memory (RAM) to enable processing of the rest of the network after the EE outputs are computed. On the other hand, if intermediate feature maps are not stored then the whole network up till the EE has to be recomputed every time, leading to significant latency increase.
> Nonetheless, the memory required to store the intermediate feature maps is nominal. Appendix C has more details.
>
> > In the right column of line 303, the authors state: “On Big-MCU, QUTE achieves 31% and 47% latency reductions over EE-ensemble and DEEP [...]
>
> This question can be broken down into three sub-questions that we answer here.
>
>
> *Why EE-ensemble has more memory overhead compared to DEEP?*
>
> As described in Section 3 and Appendix B.5.1, EE-ensemble method requires additional fully-connected (FC) layers at each EE to ensure that the learning capacities of all EEs closely match that of the final exit. Failure to do so will result in suboptimal results (Table 8). However, the FC layers are very parameter-heavy resulting in a significant increase in model parameters and thus model size. Furthermore, in the small models we evaluate (e.g., DSCNN, Resnet-8), which have only a few thousand parameters, the memory overhead of the FC layers of EE-ensemble is significantly greater compared to the base network size.
>
> *Why disparity in latency reduction?*
>
> We use two ensemble members for DSCNN and Resnet-8.  For DEEP, the MCU has no parallel computation capability, hence, each ensemble member has to be computed sequentially. Therefore, for two ensemble members, the latency the doubles. On the other hand, the latency for EE-ensemble is less compared to DEEP despite having higher memory overhead because EE-ensemble only has the computation overhead of a FC layer at each early-exit. Therefore single-forward-pass methods like EE-ensemble (and QUTE) are always more compute efficient compared to DEEP, and this latency gap will become more prominent for larger ensemble sizes. Appendix C provides a more detailed analysis.
>
> *If computed sequentially, shouldn't the memory consumption of DEEP be lower?*
>
> The memory numbers (size) reported in Figure3 is the *code-size* that reflects the amount of flash memory occupied by the program's code (instructions) and data (model parameters). This differs from RAM, which stores intermediate variables during execution. In DEEP, ensemble members are computed sequentially, avoiding extra storage for intermediate feature maps. Consequently, DEEP's peak RAM usage is consistently 4% lower than EE-ensemble (reported in Tables 10 & 11, Appendix C). However, *code-size* remains unchanged regardless of execution mode, as each ensemble member requires unique model parameters to be stored in the flash memory.
>
> [1] Rahaman et al., 2021: Uncertainty Quantification and Deep Ensembles NeurIPS 2021

---

> > ### Comment · Reviewer_eK42 · 2025-04-07
> >
> > Thank you for the clarification. I have no further concerns and will maintain the current score.

---

> > > ### Author Response · Authors · 2025-04-07
> > >
> > > Thank you for the response and for your thoughtful review. We sincerely appreciate the time and effort you dedicated to evaluating our submission.

---

### Official Review · Reviewer_XBf5 · 2025-03-15

**Overall Recommendation:** 4

**Summary:**

In this paper, the authors propose a novel method for uncertainty estimation in low-resource configurations. Specifically, the method builds on the well-known early-exit approach—where the model produces multiple predictions as the prediction depth increases and then combines them in an ensemble-like manner—but modifies it to be more efficient for TinyML applications while maintaining high uncertainty quality. To achieve this, the authors train the early-exit blocks with a set of ensemble heads applied at the end of the model. However, during inference, they completely discard the early exits, retaining only the ensemble heads while reusing the early-exit blocks' weights for these ensembles. The authors compare their method against several popular uncertainty estimation approaches on widely used benchmarks and demonstrate that it provides high-quality uncertainty estimates while being significantly faster than other alternatives.

**Claims And Evidence:**

The authors make several claims about the proposed method, including increased efficiency (in terms of both latency and model size), improved calibration, and higher uncertainty quality of the predictions. They adequately justify these claims by discussing their motivation and supporting them with experimental results in the experimental section, along with extensive ablation studies.

**Essential References Not Discussed:**

No essential references are missing.

**Experimental Designs Or Analyses:**

The experimental design adequately covers various aspects of the proposed approach, including in-distribution prediction accuracy, generalization to corrupted data, calibration, and out-of-distribution detection quality. The rigorous ablation studies are highly appreciated and provide strong support for the method's claims. Furthermore, the comparison with several popular uncertainty estimation methods further reinforces the case for its effectiveness.

**Methods And Evaluation Criteria:**

The choice of baselines (ensemble methods) and benchmarks (MNIST, CIFAR, TinyImageNet, etc.) is appropriate and accurately reflects the current state of research in this area. The evaluations are rigorous and effectively demonstrate the approach's effectiveness from multiple perspectives.

**Other Comments Or Suggestions:**

N/A

**Other Strengths And Weaknesses:**

In short, "Strengths" of the paper are:

* The proposed method adapts early-exit strategies to improve efficiency in TinyML applications while maintaining high uncertainty quality. By reusing early-exit block weights for ensemble heads, it significantly reduces computational overhead during inference. It's an original idea which has interesting application and significance.

* The paper thoroughly evaluates the method on widely used benchmarks, including MNIST, CIFAR, and TinyImageNet. Extensive ablation studies further support the claims, demonstrating strong performance across multiple uncertainty estimation tasks. Though, experiments on larger datasets would be also appreciated.

* The method is specifically designed for edge applications where uncertainty estimation might essential (for example, AV context) but computational resources are limited. Its efficiency and effectiveness against standard baselines might make it a valuable contribution to real-world deployment scenarios.

**Questions For Authors:**

N/A

**Relation To Broader Scientific Literature:**

The paper positions itself as a contribution specifically for low-latency edge applications, where uncertainty estimation is crucial despite limited computational resources. While the idea of early exits has been explored in the uncertainty estimation literature—for example, by Antoran et al. (2020)—the proposed method refines this concept to make it suitable for low-resource settings, which is one of its key contributions. In addition to demonstrating its efficiency, the authors also show its effectiveness compared to standard baselines in the field of uncertainty estimation.

**Theoretical Claims:**

No theoretical claims, theorems, or proofs are presented in the paper.

---

> ### Author Rebuttal · Authors · 2025-03-29
>
> Thank you for your detailed and insightful comments. We are glad to hear that you appreciate the novelty of our approach and recognize the thorough evaluation of the proposed method.
>
> > experiments on larger datasets would be also appreciated
>
> To demonstrate the scalability of our proposed method, we apply QUTE to a traditional Resnet50, a much larger model with higher learning capacity. We train it on TinyImagenet for 50 epochs with a batch size of 128. The rest of the training methodology is the same as described in the paper.
> We evaluate its accuracy-drop detection capability on TinyImagenet-corrupted. We report the average AUPRC over all corruptions for five severity levels below.
>
> | Method | Sev-1 | Sev-2 | Sev-3 | Sev-4 | Sev-5 |
> | :---: | :---: |  :---: | :---: |  :---: |  :---:  |
> | BASE | 0.26 | 0.54 |  0.70 | 0.67 |  0.61 |
> | EE-ensemble |  0.32 | 0.60 |  0.73 | 0.67 | 0.66 |
> | QUTE | **0.39** | **0.63** |  **0.79** | **0.75** | **0.79** |
>
> As seen, QUTE comfortably outperforms EE-ensemble and BASE on accuracy-drop detection, especially showing a remarkable 19% improvement at the highest severity level compared to EE-ensemble. This illustrates QUTE's superior accuracy-drop detection capabilities across datasets/models of varying complexities, providing a highly-effective and cost-efficient accuracy monitoring mechanism. We will include this result in the final version of the paper.

---

> > ### Comment · Reviewer_XBf5 · 2025-04-03
> >
> > Thank you for your thoughtful response to the reviews. I appreciate the clarifications and the additional experiments. I find this paper to be a valuable contribution to the area of low-resource uncertainty-based models and will therefore keep my original rating.

---

> > > ### Author Response · Authors · 2025-04-04
> > >
> > > We are truly grateful for your detailed review and for acknowledging our work’s contributions. We appreciate the time and effort you've put into evaluating our work, and your recognition means a lot.

---

### Decision · Program_Chairs · 2025-05-01

**Decision:**

Accept (poster)

**Comment:**

This manuscript received favourable reviews from all 4 reviewers. The reviews and discussions highlighted both strengths and weaknesses but the general tendency is in favour.

On the side of strengths, it was commented that the paper proposed a method which adapts early-exit strategies to improve efficiency in TinyML applications while maintaining high uncertainty quality. By reusing early-exit block weights for ensemble heads, it significantly reduces computational overhead during inference. It's an original idea which has interesting application and significance. The paper thoroughly evaluates the method on widely used benchmarks, including MNIST, CIFAR, and TinyImageNet. Extensive ablation studies further support the claims, demonstrating strong performance across multiple uncertainty estimation tasks. The method is specifically designed for edge applications where uncertainty estimation might essential (for example, AV context) but computational resources are limited. Its efficiency and effectiveness against standard baselines might make it a valuable contribution to real-world deployment scenarios.

On the side of weaknesses, it was commented that experimental evaluations for failure prediction and accuracy-drop detection are conducted based on single runs without reporting repeated trials or statistical summaries (e.g., mean and standard deviation), reducing confidence in the reliability of results, calibration experiments do not clearly demonstrate the method's superiority over baselines, as the reported advantages are minor relative to the standard deviations provided. Experiments on larger datasets would be appreciated. The paper could benefit from a more detailed discussion of recent related works to better contextualize the contributions, the paper relies mostly on experiments and does not provide a formal theoretical analysis, implying that some theoretical support would make it stronger. While the results are acceptable, more diverse real-world evaluations would strengthen the paper and further show the method’s generalizability.

The rebuttals alleviated some concerns and the reviewers reiterated their general positive evaluation of this paper.